# SELVABOX: A HIGH-RESOLUTION DATASET FOR TROPICAL TREE CROWN DETECTION

**Hugo Baudchon**[1,2,†]**, Arthur Ouaknine**[1,3,4]**, Martin Weiss**[1,2]**, Mélisande Teng**[1,2]**,
Thomas R. Walla**[5]**, Antoine Caron-Guay**[2]**, Christopher Pal**[1,6]**, Etienne Laliberté**[2,1,4]

[1]Mila – Quebec AI Institute    [2]Université de Montréal    [3]McGill University
[4]Rubisco AI    [5]Colorado Mesa University    [6]Polytechnique Montreal
[†]`hugo.baudchon@umontreal.ca`

## ABSTRACT

Detecting individual tree crowns in tropical forests is essential to study these complex and crucial ecosystems impacted by human interventions and climate change. However, tropical crowns vary widely in size, structure, and pattern and are largely overlapping and intertwined, requiring advanced remote sensing methods applied to high-resolution imagery. Despite growing interest in tropical tree crown detection, annotated datasets remain scarce, hindering robust model development. We introduce SELVABOX, the largest open-access dataset for tropical tree crown detection in high-resolution drone imagery. It spans three countries and contains more than 83 000 manually labeled crowns – an order of magnitude larger than all previous tropical forest datasets combined. Extensive benchmarks on SELVABOX reveal two key findings: (1) higher-resolution inputs consistently boost detection accuracy; and (2) models trained exclusively on SELVABOX achieve competitive zero-shot detection performance on unseen tropical tree crown datasets, matching or exceeding competing methods. Furthermore, jointly training on SELVABOX and three other datasets at resolutions from 3 to 10 cm per pixel within a unified multi-resolution pipeline yields a detector ranking first or second across all evaluated datasets. Our dataset,[1] code,[2,3] and pre-trained weights are made public.

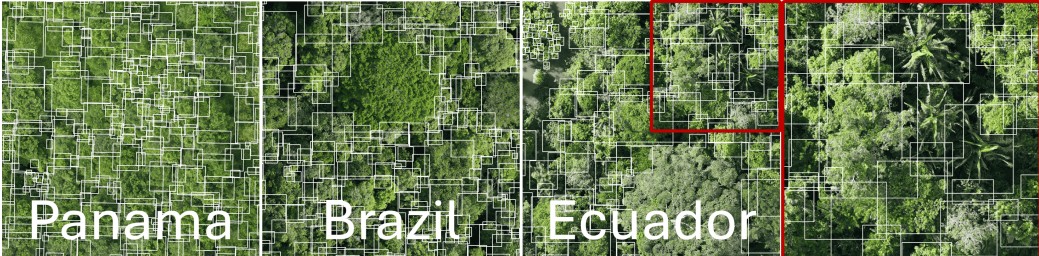

Figure 1: **The SELVABOX dataset.** The illustrated samples are extracted from rasters recorded in Panama, Brazil and Ecuador with a spatial extent of 80m × 80m and a resolution of 1.2 to 5.1 cm per pixel. The red square on the right highlights a zoom of the Ecuador sample with a spatial extent of 40m × 40m at the same resolution.

## 1 INTRODUCTION

Tropical forests cover 10% of the land area, but they store most of the biomass and biodiversity of plants on our planet (Pan et al., 2011; Gatti et al., 2022). Large trees that reach the upper canopy have a disproportionate influence on the functioning of tropical forests, with the largest 1% of trees storing

---

[1]SELVABOX dataset: `https://huggingface.co/datasets/CanopyRS/SelvaBox`
[2]Preprocessing library (*geodataset*): `https://github.com/hugobaudchon/geodataset`
[3]Inference, training & benchmark (*CanopyRS*): `https://github.com/hugobaudchon/CanopyRS`

half of the carbon of forests worldwide (Lutz et al., 2018). However, tree demography patterns in tropical forests are being altered, with increasing tree mortality, due to climate change (Brienen et al., 2015; Bonan, 2008; Esquivel-Muelbert et al., 2019) and human interventions (Harris et al., 2021). As such, monitoring individual trees in tropical forests is essential to understand the current and future potential of these forests to regulate the global climate (Davies et al., 2021).

Monitoring tropical trees is a difficult task involving slow, costly, and dangerous ground surveys by forest technicians (de Lima et al., 2022). Forest plots of tens of hectares are the gold standard of tropical tree monitoring to measure and map each individual, but completing a single one can take years of dedicated work by large teams of experts (Davies et al., 2021). Remote sensing technologies considerably augment field work, facilitating forest cartography through aerial detection of individual trees across spatial extents vastly exceeding the practical limitations of ground-based inventories (Brandt et al., 2020). Satellite imagery has been used successfully in forest monitoring tasks (Ouaknine et al., 2025), including height map estimation (Tolan et al., 2024; Lang et al., 2023) and individual tree crown detection (Brandt et al., 2020; Tucker et al., 2023; Zheng et al., 2020; 2023; 2025). However, the highest resolution satellite imagery is typically 0.3-0.5 m, which is still too coarse to distinguish trees in dense tropical forest canopies. Moreover, cloudy conditions complicate satellite sensing in the tropics.

By contrast, unoccupied aerial vehicles (UAVs) or drones can achieve cm-resolution ($< 5\,\text{cm}$), albeit at the expense of spatial coverage (Reiersen et al., 2022; Vasquez et al., 2023; Cloutier et al., 2024). Recently, datasets of UAV LiDAR (Puliti et al., 2023; 2025; Gaydon & Roche, 2025) and methods for forest structure assessment with LiDAR data have been extensively developed (Bai et al., 2023; Ma et al., 2023; Vermeer et al., 2024; Henrich et al., 2024). However, the cost of LiDAR sensing limits its adoption in tropical contexts where researchers are financially disadvantaged (de Lima et al., 2022) and calls for the development of RGB-only methods. Most open access high-quality, high-resolution tree detection RGB datasets represent temperate forests of the global North (Tab. 1). Tropical forests remain severely underrepresented and have relatively modest annotation counts (Ball et al., 2023b; Vasquez et al., 2023) despite their critical significance for biodiversity and carbon storage.

The high tree species diversity (Gatti et al., 2022) and heterogeneity in crown sizes, shapes and textures (Fig. 1 and 2) in tropical forests pose unique challenges. Indeed, solving the problem of detecting numerous objects of highly variable sizes within the same scene is still an open topic in computer vision applied to remote sensing (Rabbi et al., 2020; Li et al., 2021; Bashir & Wang, 2021). While convolutional neural networks (CNNs) remain the predominant approach for individual tree crown detection (Weinstein et al., 2019; Zamboni et al., 2021; Onishi & Ise, 2021; Yu et al., 2022; Ball et al., 2023b; Zhao et al., 2023; Bountos et al., 2025; Hajjaji et al., 2025), recent studies have explored transformer-based architectures on satellite imagery (Jiang et al., 2025), motivated by their effectiveness in multi-scale object recognition tasks (e.g., Liu et al. (2021); Zhang et al. (2023)). However, a comprehensive, resolution-aware benchmark comparing these two paradigms on UAV imagery across diverse forest ecosystems and out-of-distribution scenarios remains absent. With the growing number of UAV datasets acquired with different flight parameters, models that can generalize across resolutions and standardized frameworks are needed to bridge the gap between the ecology and computer vision communities.

We address these challenges through our contributions: ① SELVABOX, a high-resolution drone imagery dataset spanning three neotropical countries (Brazil, Ecuador, Panama) and comprising over 83 000 manual bounding box annotations on individual tree crowns; ② An exhaustive benchmark of detection methods at varying resolutions and input sizes, including a standardized evaluation framework for UAV rasters and a comprehensive assessment of model generalization on out-of-distribution (OOD) samples; ③ State-of-the-art models trained for tree crown detection out-performing competing methods on both topical and non-tropical forest datasets, in both in-distribution (ID) and OOD settings; and ④ two open-source Python libraries facilitating raster preprocessing, inference, postprocessing and standardized benchmarking. These contributions aim to simultaneously advance tropical forest monitoring and applications of machine learning to critical environmental challenges.

## 2 RELATED WORK

**Datasets.** High-resolution drone imagery enables detailed tree characterization at the pixel level (see Figure 1). This capability has catalyzed the development of open access forest monitoring

datasets (Ouaknine et al., 2025) specifically designed for tree crown semantic segmentation tasks, including pixel-wise canopy mapping (Galuszynski et al., 2022), woody invasive species identification (Kattenborn et al., 2019), and tree species classification (Cloutier et al., 2024; Kattenborn et al., 2020).

Tree crown semantic segmentation, a pixel-wise classification task, cannot inherently distinguish individual trees, making it unsuitable for applications such as tree counting or biomass estimation where individual tree crown detection and delineation are essential (Fu et al., 2024). Datasets for individual tree crown detection (Weinstein et al., 2021; Reiersen et al., 2022) and delineation (Ball et al., 2023b; Firoze et al., 2023; Vasquez et al., 2023; Cloutier et al., 2024; Lefebvre & Laliberté, 2024; Veitch-Michaelis et al., 2024), corresponding to object detection and instance segmentation tasks respectively, have been proposed

Table 1: **Related datasets.** The number of tree crowns manually[*] annotated ('# Trees') are noted in 'k' for thousands. The reported resolution or ground sampling distance ('GSD') is in centimeter per pixel. We define the forest 'type' as either urban, plantation, natural; 'biome' as either temperate, tropical or worldwide (when the dataset spans over several biomes). [*]except for ReforesTree, see Section 4.

| Name | # Trees | GSD | Type | Biome |
|---|---|---|---|---|
| NeonTreeEval. (Weinstein et al., 2021) | 16k | 10 | natural | temperate |
| ReforesTree (Reiersen et al., 2022) | 4.6k | 2 | plantation | tropical |
| Firoze *et al.* (Firoze et al., 2023) | 6.5k | 2–5 | natural | temperate |
| Detectree2 (Ball et al., 2023b) | 3.8k | 10 | natural | tropical |
| BCI50ha (Vasquez et al., 2023) | 4.7k | 4.5 | natural | tropical |
| BAMFORESTS (Troles et al., 2024) | 27k | 1.6–1.8 | natural | temperate |
| QuebecTrees (Cloutier et al., 2024) | 23k | 1.9 | natural | temperate |
| Quebec Plantation (Lefebvre & Laliberté, 2024) | 19.6k | 0.5 | plantation | temperate |
| OAM-TCD (Veitch-Michaelis et al., 2024) | 280k | 10 | mostly urban | worldwide |
| SELVABOX (ours) | **83k** | **1.2–5.1** | **natural** | **tropical** |

for both general forest monitoring and specialized applications such as dead tree identification (Mosig et al., 2024). Table 1 summarizes open access datasets for general tree crown monitoring. Despite considerable community efforts to share manually annotated tree crown data, a substantial gap remains in datasets for monitoring tropical trees in natural forests.

**Modeling.** Deep learning is the dominant paradigm for individual tree crown delineation, superseding earlier computer vision and machine learning methods (Kattenborn et al., 2021). Open access datasets (Tab. 1) have facilitated the development of individual tree crown detection models with deep learning architectures, including Faster R-CNN (Ren et al., 2015b), Mask R-CNN (He et al., 2017), and RetinaNet (Lin et al., 2017), as demonstrated with DeepForest (Weinstein et al., 2019) and Detectree2 (Ball et al., 2023b). These CNN-based methods have proven effective in diverse forest scenarios (Zhao et al., 2023). Tree crown models have also leveraged SAM (Kirillov et al., 2023) by providing efficient prompts for zero-shot tree crown delineation (Teng et al., 2025). While the FoMo benchmark (Bountos et al., 2025) has explored transformer-based architectures including pretrained DeiT (Touvron et al., 2021) and DINOv2 (Oquab et al., 2024) backbones, advanced transformer-based object detection methods (Liu et al., 2022; Zhang et al., 2023) remain underexplored in this domain.

**Evaluation.** Previous open access datasets (Tab. 1) have evaluated detection methods using classification-based metrics per tree (recall, precision, F1-score) (Weinstein et al., 2020; 2021; Zheng et al., 2021; Beloiu et al., 2023) with detection-based metrics such as intersection over union (IoU) and mean average precision (mAP) (Hao et al., 2021; Yu et al., 2022; Ball et al., 2023b; Fu et al., 2024; Firoze et al., 2023; Veitch-Michaelis et al., 2024; Bountos et al., 2025). UAV rasters are usually divided in tiles for training and evaluation, but tile-level metrics are susceptible to edge effects (where partial trees appear at tile boundaries) and duplicate detections when scaled to larger areas, complicating accurate tree counting. As a consequence, tile-level metrics fail to accurately represent performances at the entire raster level, which is what matters to practioners. For example, tracking the mortality of large tropical trees over time and across vast areas requires aggregating detections from individual tiles into a coherent raster-level map. A recall metric for keypoint-in-tree prediction tasks at the raster level was proposed to evaluate OAM-TCD (Veitch-Michaelis et al., 2024). In this work, we extend the evaluation of aggregated predictions from individual images to detection-based tasks, including both precision and recall metrics (F1-score) as well as the location of each object.

**Multi-resolution.** Despite growing interest in multi-scale and multi-resolution analysis for deep learning in remote sensing applications (Reed et al., 2023; Bountos et al., 2025), these approaches remain understudied for forest monitoring. While increased spatial extent per tile improves tree crown classification (Näsi et al., 2015; Liu et al., 2020; Kattenborn et al., 2020) and higher tile resolution benefits tree crown semantic segmentation more than increased spatial extent (Schiefer et al., 2020), resolution-induced domain shift remains challenging for individual tree crown detection.

Current pre-trained models (e.g., DeepForest, Detectree2) show poor zero-shot performance on OOD samples (Gan et al., 2023), though targeted fine-tuning can mitigate this gap (Bountos et al., 2025). Further research is needed to evaluate how tile spatial extent, size, and resolution impact detection performance and to develop fine-tuning methodologies that reduce zero-shot degradation on OOD samples, particularly given the substantial size variation in tropical tree crowns (Fig. 2).

## 3 THE SELVABOX DATASET

We present SELVABOX, a large-scale benchmark dataset addressing the critical open-access annotation scarcity in tropical forest remote sensing (Sec. 2) while motivating research in individual tree crown detection. SELVABOX encompasses 83 137 individual tree crown bounding boxes on top of 14 RGB orthomosaics, including 96.6 ha in Brazil, 96 ha in Panama and 318.1 ha in Ecuador, recorded with four different drones (DJI Mavic 3 Entreprise [m3e], DJI Mavic 3 Multispectral [m3m], DJI Mavic Pro [mavicpro], DJI Mavic Mini 2 [mini2]) at ground sampling distance

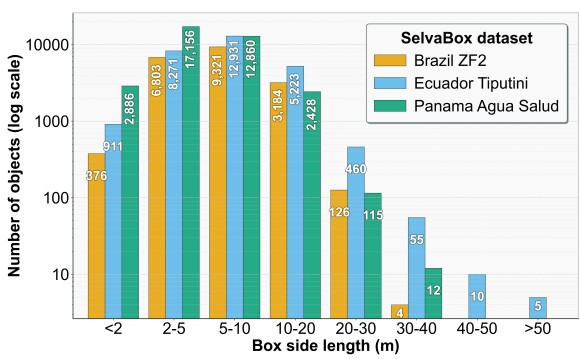

Figure 2: **Distribution of box annotations size in SELVABOX per country.**

(GSD) between 1.2–5.1 cm per pixel (Tab. 6 in App. A.1). Our drone imagery was acquired over primary and secondary forests, and native tree plantations. It includes diverse sets and shapes of tropical trees as depicted in Figure 1. More details about the orthomosaics can be found in App. A.1.

**Locations.** The RGB imagery was acquired in three countries: Brazil, Ecuador, and Panama (Tab. 6 in App. A.1). The Brazil data was collected at the ZF-2 station, a forest with high-diversity characteristic of the Central Amazon and growing on nutrient-poor soils. The topography consists of plateaus dissected by valleys (Amaral et al., 2019). The Ecuador data was recorded at the Tiputini Biodiversity Station (TBS), located within the Yasuní Biosphere Reserve, one of the most biodiverse forests on Earth (Valencia et al., 2004). The climate of this Western Amazonia region is considered to be aseasonal compared to Central Amazonia while the soils tend to be richer in nutrients as they are derived from younger sediments from the Andes (Hoorn et al., 2010). Finally, the Panama data was acquired from four areas of the Agua Salud Project (Mayoral et al., 2017). Two areas are plantations of native tree species (Mayoral et al., 2017), while the other two are from surrounding secondary forests. The soils of Agua Salud are acidic and nutrient-poor (van Breugel et al., 2019). The tree species diversity of Central Panama is considered lower than our other two Amazonian sites.

**Annotations.** The data was manually annotated by five trained biologists. They were asked to draw bounding boxes around every individual tree crown they could reliably detect from the imagery. They generated 83 137 manual tree annotations during 1 284 people-hours with crowns spanning from $< 2$ m to $> 50$ m in diameter (Fig. 2). All annotations were produced with ArcGIS Pro version 3.0, stored in hosted feature layers on ArcGIS Online, have georeferenced coordinates, and were exported to geopackages. Figure 2 shows the tree crown annotation bounding-box size distribution, where we notice a long-tail distribution for larger trees, especially in Ecuador.

Our annotation process used photo-interpretation, the most reliable and feasible approach at this scale. Field-based validation faces significant technical constraints in tropical forests, including GNSS signal blockage, multipath errors from dense canopy, difficulties linking non-straight tree trunks to canopy imagery, and variable geolocation errors—making the process less efficient and accurate than photo-interpretation (Laliberté et al., 2025). Additionally, logistical challenges like intense heat, humidity, heavy rain, and dense vegetation make fieldwork costly and hazardous. Since LiDAR data requires expensive equipment and specialized annotators compared to photo-interpretation, it is less accessible for tropical forest scientists, who have limited research funding (de Lima et al., 2022). Consequently, we adopted an RGB-only validation approach to ensure scalability and broad applicability. The annotation process followed a standardized protocol with initial training of domain-expert annotators, multi-pass annotation reviews, and systematic quality control (see App. A.3 for

full details). We supplemented visual interpretation with digital surface models (DSMs) derived from 3D photogrammetric point clouds, using elevation data to distinguish adjacent crowns with similar visual features but different heights.

**Spatially separated splits.**  We propose train, validation and test splits, created spatially in the rasters to ensure no pixel overlap between splits and avoid geospatial auto-correlation (Kattenborn et al., 2022), and including 61.4k, 9.6k, and 10.6k boxes respectively. We define our splits by manually creating areas of interest (AOIs) geopackages in the QGIS software (Fig. 5 in App. A.2). Orthomosaic borders with poor visual quality were deliberately excluded during AOI creation to ensure clean, artifact-free splits. For the test split, we defined the AOIs on rasters with minimal visual reconstruction artifacts while including a maximal diversity and quality in box annotations.

**Incomplete annotations.**  Although considerable effort was put into producing a dense tree-crown mapping during the annotation process, some annotators reported difficulties clearly distinguishing a subset of individual trees on one raster in Brazil and three rasters in Ecuador, resulting in sparser annotations. Annotation sparsity is a common challenge in tree detection datasets: The Detectree2 dataset contains only tiles covered in area by at least 40% tree crown annotation polygons (Ball et al., 2023b). This method introduces noise during the training process as annotations may be missing for up to half of the trees in an image, causing misleading penalization. We adopt a different strategy where we mask targeted pixels in our AOIs with missing annotations when dividing the rasters in tiles. During training, we expect models to become agnostic to such masked pixels, *i.e.* not predicting boxes in those areas, thus not being penalized due to missing annotations. Such holes were created for train AOIs, a sub-set of valid AOIs, while test AOIs were chosen to cover areas where annotations are dense and complete. Figure 6 (in App. A.4) shows an example of pixels masked that way.

**Tiling and preprocessing.**  When tiling the rasters, *i.e.* dividing rasters into tiles, we use AOI geopackages to mask pixels that are outside of each tile's assigned split. Each tree crown annotation is assigned to a single split where it overlaps the most according to the AOIs. For each tile, we keep annotations that overlap at least at 40% with the tile's extent. For the ready-to-train dataset, we remove tiles that contain no annotations, more than 80% black (masked), white or transparent pixels. A sliding-window tiling approach was used, with 50% tile overlap for the training and validation splits, and 75% for the test split to ensure that the largest trees entirely fit in at least one tile (Sec. 4). We release our preprocessing pipeline as a python library called *geodataset*. The final preprocessed dataset is available on HuggingFace under the permissive CC-BY-4.0 license.

## 4    BENCHMARKING MODELS AND METHODS

We structure our experiments in three sequential phases. First, we identify effective modeling choices by evaluating various object detection models and input image settings on SELVABOX, examining how resolution and spatial extent influence detection accuracy based on in-distribution performance (Sec. 4.1). Second, we validate the efficacy of multi-resolution domain augmentation by testing whether multi-resolution training improves or degrades performance compared to single-resolution training (Sec. 4.1). Finally, we assess generalization to other datasets by evaluating three categories: models trained exclusively on SELVABOX, models trained on SELVABOX combined with additional datasets, and models trained without SELVABOX including external methods (Sec. 4.2).

In addition to SELVABOX, we use the OAM-TCD (Veitch-Michaelis et al., 2024), NeonTreeEvaluation (Weinstein et al., 2022; 2021), QuebecTrees (Cloutier et al., 2023; 2024), BCI50ha (Vasquez et al., 2023), and Detectree2 (Ball et al., 2023a) datasets. We excluded the Quebec Plantations dataset (Lefebvre & Laliberté, 2024), as it comprises non-tropical, young tree plantations outside the scope of our study. Similarly, we excluded ReforesTree (Reiersen et al., 2022), a tropical plantation dataset whose bounding box annotations were generated by inference from a fine-tuned DeepForest model (Weinstein et al., 2020), resulting in noisy annotations unsuitable for robust training or evaluation (Fig. 13 in App. F.4). Additionally, we omitted the dataset published by Firoze *et al.* (Firoze et al., 2023), as it was designed for image sequence-based tree detection, with annotations derived from highly overlapping, video-like image sequences, introducing redundancy and requiring extensive preprocessing. Given that each dataset varies in ground sampling distance (GSD), tree crown size distribution, annotation type, and predefined splits or areas of interest (AOIs), we applied independent

preprocessing procedures detailed in Appendix F.1. Our benchmarking, inference, and training pipelines are publicly available in our Python repository *CanopyRS*.

**Evaluation metrics.** To evaluate models at the tile level, we consider the industry-standard COCO-style $mAP_{50:95}$ and $mAR_{50:95}$ metrics (Lin et al., 2014). Due to the high number of objects per tile in SELVABOX (at 80m ground extent, see Sec. 4.1), QuebecTrees and BCI50ha, we increase the maxDets parameter of COCOEval from 100 to 400 for those datasets.

As detailed in Section 2, tile-level metrics do not necessarily reflect raster-level performance, which is the operational target for concrete applications such as large-scale forest inventories. To address this, we propose $RF1_{75}$, a Raster-level F1 score evaluating final predictions after tile aggregation via Non-Maximum Suppression (NMS). It uses the same greedy matching as COCO metrics, but requires a single, strict IoU threshold of $\geq 75\%$ for a match. This 75% threshold is a balanced criterion for dense canopies, where 50% IoU is too permissive and 90% is overly difficult. By integrating the F1 score at the raster level with this IoU restriction, the $RF1_{75}$ metric accounts for precision and recall, both important in forest monitoring applications. For each dataset, we tune NMS hyperparameters on the validation set, apply the optimal settings to the test set, and report the final $RF1_{75}$ as a weighted average over all rasters (details in App. B.3). While annotation noise makes a perfect score of 1.0 unlikely, maximizing $RF1_{75}$ is a practical target for reliable ecological monitoring.

**Model architectures and training.** We compare four object detection approaches for tree crown delineation: ① Faster R-CNN with ResNet-50 backbone (Ren et al., 2015a; He et al., 2016), a widely used CNN-based detector; ② DeepForest (Weinstein et al., 2019; 2020), a RetinaNet variant trained on NeonTreeEvaluation; ③ Detectree2 (Ball et al., 2023b), a Mask R-CNN trained on a dataset also called Detectree2, evaluated in two variants: 'resize' (multi-resolution tropical) and 'flexi' (joint tropical-urban training); and ④ DINO (Zhang et al., 2023), a DETR-based transformer model that we evaluate with both ResNet-50 and Swin-L backbones (Liu et al., 2021). Note that DINO (the DETR-based detector) and DINO (the self-supervised embedding model) are unrelated despite sharing the same name. While recent DETR-based architectures have reached similar or better performances (Zong et al., 2023), we chose DINO for its adoption by the community through Detectron2 (Wu et al., 2019) and Detrex (Ren et al., 2023). DINO, Faster R-CNN, DeepForest, and Detectree2 serve as strong and diverse baselines from both general-purpose and domain-specific tree crown detection literature (Sec. 2). All models are initialized from COCO-pretrained checkpoints. We implemented our own augmentation pipeline, and use standard crop, resize, flip, rotation and color augmentations (App. B.1). Training sessions took between 12 hours and 3 days for both architectures. All hyperparameters used for training and testing are detailed in Appendix B.2.

### 4.1 MODEL, RESOLUTION AND SPATIAL EXTENT SELECTION ON SELVABOX

We choose a raster tiling scheme that balances detection accuracy, object coverage, and hardware constraints. Our standard tile is $80 \times 80$ m at 4.5 cm/px ($1777 \times 1777$ pixels). This setting ensures that the largest crowns in SELVABOX, some upwards of 50 m in diameter (Fig. 2), fit entirely within one tile when using a 75% overlap between tiles in our test set, while keeping our models (*e.g.*, DINO 5-scale with Swin-L) trainable on 48 GB GPUs with a batch size of one per GPU.

To assess the trade-offs between spatial resolution and ground extent, we conduct an ablation study across three configurations (Sec. 5 and Tab. 3). We vary the resolution between 4.5, 6, and 10 cm/px, yielding $1777 \times 1777$, $1333 \times 1333$, and $800 \times 800$ pixel inputs for a fixed $80 \times 80$ m ground extent. In parallel, we test $40 \times 40$ m tiles, which contain fewer crowns per image and still guarantee that over 99.9% of crowns—those smaller than 30 m—are fully visible in at least one tile, assuming a 75% overlap. This ablation isolates the effects of spatial detail, object count, and input size. Each model is trained at a fixed resolution, with only minor cropping augmentation ($\pm 10\%$ of input size) before resizing to a fixed input size. Further experimental details are provided in App. C.

We also compare models trained at 6 cm and 10 cm GSD while resizing the inputs to assess the impact of both the resolution and input size on models performance. Tile-level evaluation metrics ($mAP_{50:95}$ and $mAR_{50:95}$) are not comparable *per se* between $40 \times 40$ and $80 \times 80$ m spatial extent since the tiles differ in object count and spatial boundaries. But one may compare all results with the $RF1_{75}$ since it is computed at the raster level, after aggregation of individual images predictions.

Tables 3 and 4: **Model, resolution and spatial extent selection on SELVABOX.** Comparison of performances on the proposed test set of SELVABOX with variable tile spatial extent, respectively $40 \times 40$ m in Tab. 2 and $80 \times 80$ m in Tab. 3, input size in pixels and ground spatial distance (GSD) in cm. We highlight results per method and backbone as ▢ the first, ▢ the second and ▢ the third best scores. We also **bold** and underline the best and second best scores overall. Note that $mAP_{50:95}$ and $mAR_{50:95}$ cannot be compared between $40 \times 40$ m and $80 \times 80$ m inputs as images do not match, but we can use $RF1_{75}$ to compare final post-aggregation results at the raster-level.

Table 2: SELVABOX at $40 \times 40$ m.

| Method | GSD | I. size | $mAP_{50:95}$ | $mAR_{50:95}$ | $RF1_{75}$ |
|---|---|---|---|---|---|
| Faster R-CNN ResNet50 | 10 | 400 | 26.90 (±0.13) | 40.87 (±0.35) | 35.78 (±0.44) |
| | 10 | 666 | 28.40 (±0.13) | 42.79 (±0.19) | 37.75 (±0.30) |
| | 10 | 888 | 28.51 (±0.20) | 43.36 (±0.19) | 37.46 (±0.91) |
| | 6 | 666 | 29.31 (±0.05) | 43.59 (±0.20) | 39.97 (±0.33) |
| | 6 | 888 | 29.40 (±0.34) | 44.18 (±0.44) | 38.92 (±0.51) |
| | 4.5 | 888 | 30.25 (±0.24) | 45.18 (±0.30) | 39.97 (±0.67) |
| DINO 4-scale ResNet50 | 10 | 400 | 30.63 (±0.24) | 48.06 (±0.33) | 41.14 (±0.80) |
| | 10 | 666 | 31.76 (±0.86) | 50.40 (±0.55) | 41.57 (±1.94) |
| | 10 | 888 | 32.19 (±0.33) | 50.68 (±0.19) | 42.47 (±0.97) |
| | 6 | 666 | 33.46 (±0.22) | 51.80 (±0.31) | 44.55 (±0.18) |
| | 6 | 888 | 33.54 (±0.24) | 52.12 (±0.18) | 43.34 (±0.79) |
| | 4.5 | 888 | 34.19 (±0.13) | 52.53 (±0.40) | 44.26 (±0.83) |
| DINO 5-scale Swin L-384 | 10 | 400 | 33.84 (±0.20) | 52.02 (±0.25) | 45.37 (±0.23) |
| | 10 | 666 | 34.64 (±0.25) | 52.91 (±0.30) | 46.39 (±0.52) |
| | 10 | 888 | 34.92 (±0.34) | 53.23 (±0.14) | 45.22 (±0.70) |
| | 6 | 666 | 37.07 (±0.16) | 55.18 (±0.22) | 48.50 (±0.60) |
| | 6 | 888 | 36.22 (±0.38) | 54.55 (±0.43) | 48.13 (±0.60) |
| | 4.5 | 888 | **37.78** (±0.15) | **56.30** (±0.21) | **49.76** (±0.43) |

Table 3: SELVABOX at $80 \times 80$ m.

| Method | GSD | I. size | $mAP_{50:95}$ | $mAR_{50:95}$ | $RF1_{75}$ |
|---|---|---|---|---|---|
| Faster R-CNN ResNet50 | 10 | 800 | 24.94 (±0.34) | 35.93 (±0.55) | 34.66 (±0.97) |
| | 10 | 1333 | 26.25 (±0.14) | 38.59 (±0.41) | 36.09 (±0.51) |
| | 10 | 1777 | 27.58 (±0.24) | 40.21 (±0.38) | 35.74 (±1.26) |
| | 6 | 1333 | 26.52 (±0.80) | 39.55 (±0.75) | 36.22 (±1.45) |
| | 6 | 1777 | 27.89 (±0.35) | 41.02 (±0.69) | 35.94 (±0.84) |
| | 4.5 | 1777 | 28.74 (±0.44) | 41.27 (±0.59) | 37.52 (±0.58) |
| DINO 4-scale ResNet50 | 10 | 800 | 30.90 (±0.51) | 47.29 (±0.33) | 41.20 (±0.39) |
| | 10 | 1333 | 32.39 (±0.02) | 49.22 (±0.10) | 43.08 (±0.20) |
| | 10 | 1777 | 32.51 (±0.89) | 49.35 (±0.47) | 42.39 (±1.25) |
| | 6 | 1333 | 33.06 (±0.29) | 49.93 (±0.39) | 42.92 (±0.51) |
| | 6 | 1777 | 33.62 (±0.10) | 50.85 (±0.17) | 44.18 (±0.18) |
| | 4.5 | 1777 | 33.81 (±0.84) | 51.00 (±0.77) | 43.26 (±0.45) |
| DINO 5-scale Swin L-384 | 10 | 800 | 33.90 (±0.09) | 50.29 (±0.38) | 44.64 (±0.20) |
| | 10 | 1333 | 34.22 (±0.34) | 50.76 (±0.57) | 45.64 (±1.03) |
| | 10 | 1777 | 35.30 (±0.26) | 52.12 (±0.62) | 45.37 (±0.08) |
| | 6 | 1333 | 37.12 (±0.38) | 53.56 (±0.48) | 47.81 (±0.40) |
| | 6 | 1777 | 35.77 (±0.84) | 52.91 (±0.56) | 45.88 (±1.97) |
| | 4.5 | 1777 | **37.79** (±0.55) | 54.66 (±0.47) | 49.38 (±0.76) |

**Multi-resolution approach.** Diversity in camera sensors and recording conditions results in datasets with various resolutions (Tab. 1 and 6), complicating or preventing model training across multiple datasets. We mitigate this through multi-resolution input augmentation that enforces scale-invariance during training, enabling us to combine datasets of various resolutions. This simple, yet efficient process randomly crops inputs using a wide range of crop sizes, then randomly resizes the crops. This achieves two effects: ① cropping performs ground extent augmentation, and ② resizing performs GSD augmentation. Details on our multi-resolution augmentation pipeline are in App. D.1.

While data augmentation generally improves generalization, extreme transformations may impact convergence and performance. Therefore, we train multi-resolution models on SELVABOX with increasingly large crop ranges (Fig. 3) and the same random resize in the $[1024, 1777]$ pixel range, comparing them at $80 \times 80$ m to the best single-resolution, single-input-size models from the previous experiment (*i.e.* DINO Swin-384 at 4.5, 6 and 10 cm; see Tab. 3).

## 4.2 METHODOLOGY TO EVALUATE OOD GENERALIZATION

To evaluate the generalization capabilities of models trained on SELVABOX, we define BCI50ha and Detectree2 (Tab. 1) as OOD datasets for test-only evaluation. We perform zero-shot evaluations on these datasets, meaning models are tested without any fine-tuning on data completely excluded from training, and characterized by diverse resolutions, image quality, and forest types. These two datasets are considered OOD relative to SELVABOX because ① BCI50ha is located on an island in Panama (whereas SELVABOX is on mainland Panama), and Detectree2 is located in Malaysia, on a different continent; and ② both datasets were acquired using different drones, camera sensors, and flight conditions. Additionally, we include NeonTreeEvaluation, QuebecTrees, and OAM-TCD as either in-distribution or OOD datasets to assess how varying the number and diversity of datasets used during training affects model generalization.

We compare a multi-resolution model trained exclusively on SELVABOX, using a crop augmentation range of $[30, 120]$ meters (equivalent to $[666, 2666]$ pixels), against models trained on different combinations of OAM-TCD, NeonTreeEvaluation, QuebecTrees, and SELVABOX datasets (including DeepForest and Detectree2). We selected this multi-resolution augmentation range based on our benchmark results (Sec. 5, Fig. 3), which indicated that this range achieves performance comparable to single-resolution and less aggressive multi-resolution methods on SELVABOX, while also allowing spatial extents of images from different datasets to partially overlap (Tab. 19 in App. F.1). Finally, we optimize non-maximum suppression (NMS) hyperparameters using the validation sets of SELVABOX and Detectree2, while keeping BCI50ha strictly zero-shot.

# 5 EXPERIMENTS AND RESULTS

First, we evaluate model architectures, resolutions, and spatial extents on SelvaBox (Sec. 5.1). Then, we validate our multi-resolution training methodology. Finally, we assess generalization on OOD datasets (Sec. 5.2).

## 5.1 SELVABOX RESULTS

Using the methodology in Section 4.1, we find:

**Resolution matters, transformers too.** In Tables 2 and 3, we find that for all GSD and spatial extents, DINO outperforms Faster R-CNN, and Swin L-384 outperforms ResNet-50. We also observe significant improvements in $mAP_{50:95}$, $mAR_{50:95}$ and $RF1_{75}$ when using lower GSD for all architectures. While larger input sizes at fixed resolution benefits ResNet-50-based methods, DINO + Swin L-384 models do not see such improvements at 6 cm per pixel. This suggests diminishing returns from further

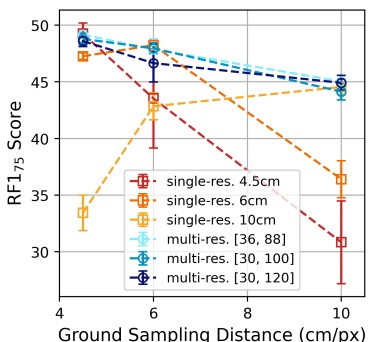

Figure 3: **Multi-resolution vs. single-resolution on SELVABOX.** $RF1_{75}$ for the best single-resolution methods from Tab. 3 trained at fixed $80 \times 80$ m extent vs multi-resolution approaches with varying crop augmentation ranges $[36, 88]$, $[30, 100]$, $[30, 120]$. All methods are 'DINO 5-scale Swin L-384'.

increases in input size, and only the Swin L-384 backbone fully leverages more detailed inputs. Finally, we observe that Faster R-CNN reaches best $RF1_{75}$ performance at $40 \times 40$ m rather than $80 \times 80$ m, likely due to larger context and higher number of objects making the task more difficult.

**Multi-resolution is effective on SELVABOX.** In Figure 3, we observe that all multi-resolution models achieve $RF1_{75}$ results within standard-deviation of the best single-resolution models, for all three resolutions. Additionally, single-resolution models struggle at test-time on unseen resolutions. Results for $mAP_{50:95}$ and $mAR_{50:95}$ are similar and presented in Appendix (Fig. 8). This demonstrates that a single multi-resolution model can be trained for transferability across spatial extents and GSDs without performance losses on SELVABOX, instead of training multiple resolution-specific models.

## 5.2 OOD RESULTS

Following the methodology described in Sec. 4.2, we evaluate zero-shot generalization, we find:

**SELVABOX exposes the limitations of current methods and datasets.** We report results on tropical forests in Table 4. First, existing methods, namely Detectree2 and DeepForest, perform poorly on SELVABOX in zero-shot evaluation with 6.08 and 13.14 $RF1_{75}$ respectively. Our method trained with multi-resolutions on NeonTreeEvaluation, QuebecTrees and OAM-TCD reaches 30.81 $RF1_{75}$ on SELVABOX in zero-shot evaluation, showing great generalization performances on unseen tropical forests. When SELVABOX is included in-distribution of the training process, our methods achieve state-of-the-art performances with 47.63 (multi-datasets + SELVABOX) and 48.60 (SELVABOX only) $RF1_{75}$. These experimental results show how challenging SELVABOX is for existing methods, filling a gap not covered by existing datasets and methods.

**SELVABOX improves OOD generalization on tropical datasets.** We observe that models trained on SELVABOX achieve state-of-the-art performance in zero-shot evaluation on BCI50ha, at 39.39 (multi-datasets + SELVABOX) and 41.91 (SELVABOX only) $RF1_{75}$, followed by Detectree2-resize at 34.97 $RF1_{75}$. On the Detectree2 dataset, the best performing model is Detectree2-resize in $RF1_{75}$ although a potential data leak could have occurred during the evaluation on their dataset, which limits the interpretation of the results, given that we were unable to recover the training-test splits originally used. Our multi-dataset + SELVABOX method outperforms both Detectree2's models in terms of $mAP_{50:95}$ and $mAR_{50:95}$ on the Detectree2 dataset and beats DeepForest. It also outperforms our multi-dataset without SELVABOX and SELVABOX-only methods, while being evaluated on a restricted zero-shot regime. We include corresponding qualitative results in Appendix F.5. To our knowledge, the DINO-Swin-L trained on multi-dataset + SELVABOX including a multi-resolution training process achieves state-of-the-art performance for the tropical tree crown detection task, generalizing well on both SELVABOX and OOD tropical datasets.

Table 4: **Tropical datasets evaluation**. We respectively denote N for NeonTreeEvaluation, D for Detectree2, D+u for Detectree2 with urban regions, Q for QuebecTrees, O for OAM-TCD, S for SELVABOX and B for BCI50ha. We noted OD to identify out-of-distribution datasets, and RG for relative gain in $RF1_{75}$ of each method compared to the best competing one (Detectree2-rezise in this table). We mark the best and second-best scores in **bold** and underline, respectively. We denote with $\sim$ the Detectree2 competing methods where original train-test splits could not be recovered, preventing controlled evaluation on their dataset and limiting the interpretability of comparative results. Standard deviations (over three seeds) are reported only for models we trained ourselves, whereas Detectree2 and DeepForest (N) rely on single released model without variability estimates.

| Method | Train set(s) | SELVABOX (S) $mAP_{50:95}$ | $mAR_{50:95}$ | $RF1_{75}$ | OD | RG | Detectree2 (D) $mAP_{50:95}$ | $mAR_{50:95}$ | $RF1_{75}$ | OD | RG | BCI50ha (B) $mAP_{50:95}$ | $mAR_{50:95}$ | $RF1_{75}$ | OD | RG |
|---|---|---|---|---|---|---|---|---|---|---|---|---|---|---|---|---|
| Detectree2-resize | D | 8.62 | 15.47 | 13.14 | ✓ | 0% | 17.67 | 34.11 | **23.87** | ~ | **0%** | 32.11 | 48.18 | 34.97 | ✓ | 0% |
| Detectree2-flexi | D+u | 6.43 | 13.20 | 9.21 | ✓ | -30% | 6.43 | 19.86 | 4.46 | ~ | -82% | 12.72 | 29.47 | 4.26 | ✓ | -88% |
| DeepForest | N | 4.70 | 9.08 | 6.08 | ✓ | -54% | 6.85 | 19.27 | 7.83 | ✓ | -68% | 14.48 | 25.50 | 10.02 | ✓ | -72% |
| F. R-CNN-RN50 | N | 1.79(±0.21) | 11.08(±0.01) | 4.54(±0.33) | ✓ | -66% | 11.09(±1.58) | 26.28(±2.38) | 14.80(±2.57) | ✓ | -38% | 0.72(±0.12) | 4.47(±0.95) | 1.42(±0.18) | ✓ | -96% |
| DINO-Swin-L | N | 5.67(±0.73) | 17.63(±1.13) | 9.94(±2.12) | ✓ | -25% | 14.77(±3.58) | 32.62(±4.06) | 19.87(±4.12) | ✓ | -17% | 1.74(±0.35) | 11.89(±0.51) | 3.77(±0.59) | ✓ | -90% |
| DeepForest | S | 28.84(±0.19) | 44.67(±0.09) | 38.00(±0.22) | ✗ | +189% | 6.34(±1.11) | 18.35(±1.78) | 2.71(±0.67) | ✓ | -89% | 25.17(±1.09) | 46.85(±0.71) | 36.46(±1.38) | ✓ | +4% |
| F. R-CNN-RN50 | S | 28.49(±0.05) | 41.88(±0.25) | 36.37(±0.37) | ✗ | +176% | 3.32(±0.76) | 11.50(±1.31) | 1.04(±0.66) | ✓ | -96% | 27.23(±1.49) | 46.70(±1.64) | 31.24(±1.39) | ✓ | -11% |
| DINO-Swin-L | S | **37.77(±0.35)** | **54.69(±0.07)** | **48.60(±0.49)** | ✗ | +269% | 13.27(±1.80) | 28.24(±2.75) | 8.47(±3.13) | ✓ | -65% | **36.87(±0.67)** | **60.30(±0.90)** | **41.91(±1.28)** | ✓ | +19% |
| DeepForest | NQO | 14.93(±1.23) | 31.76(±1.05) | 21.55(±1.57) | ✓ | +64% | 10.96(±1.78) | 26.14(±2.20) | 8.19(±2.46) | ✓ | -66% | 10.84(±0.94) | 31.13(±2.08) | 18.58(±1.10) | ✓ | -47% |
| F. R-CNN-RN50 | NQO | 16.39(±0.11) | 29.39(±0.11) | 24.77(±0.38) | ✓ | +88% | 12.50(±0.42) | 28.17(±0.64) | 13.65(±0.92) | ✓ | -43% | 11.92(±3.43) | 32.74(±4.29) | 16.16(±3.36) | ✓ | -54% |
| DINO-Swin-L | NQO | 20.85(±1.46) | 39.87(±1.66) | 30.81(±1.53) | ✓ | +134% | 15.35(±1.88) | 30.51(±2.72) | 11.31(±2.55) | ✓ | -53% | 25.72(±1.92) | 48.78(±1.72) | 25.32(±1.87) | ✓ | -28% |
| DeepForest | NQOS | 27.58(±0.54) | 43.69(±0.44) | 35.92(±1.20) | ✗ | +173% | 12.77(±0.31) | 29.39(±0.36) | 9.13(±0.45) | ✓ | -20% | 19.53(±1.92) | 43.52(±3.19) | 28.00(±3.81) | ✓ | -20% |
| F. R-CNN-RN50 | NQOS | 24.93(±1.10) | 39.34(±0.38) | 30.56(±1.44) | ✗ | +132% | 13.80(±1.91) | 29.84(±2.79) | 14.42(±2.69) | ✓ | -40% | 20.42(±1.48) | 43.25(±1.56) | 23.49(±1.17) | ✓ | -33% |
| DINO-Swin-L | NQOS | 36.95(±0.56) | 53.71(±0.32) | 47.63(±0.23) | ✓ | +262% | **18.20(±3.22)** | **35.20(±3.61)** | 19.23(±3.33) | ✓ | -20% | 33.13(±3.06) | 58.36(±2.21) | 39.39(±1.71) | ✓ | +12% |

Table 5: **Non-tropical datasets evaluation**. We respectively denote N for NeonTreeEvaluation, D for Detectree2, D+u for Detectree2 with urban regions, Q for QuebecTrees, O for OAM-TCD, S for SELVABOX and B for BCI50ha. We noted OD to identify out-of-distribution datasets, and RG for relative gain in $RF1_{75}$ if available, $mAP_{50:95}$ otherwise, of each method compared to the best competing one (either DeepForest (N) or Detectree2-flexi in this table). We mark the best and second-best scores in **bold** and underline, respectively. We cannot compute $RF1_{75}$ for NeonTreeEvaluation and OAM-TCD as only individual images are available for their test splits. Standard deviations (over three seeds) are reported only for models we trained ourselves, whereas Detectree2 and DeepForest (N) rely on single released model without variability estimates.

| Method | Train set(s) | NeonTreeEvaluation (N) $mAP_{50:95}$ | $mAR_{50:95}$ | $RF1_{75}$ | OD | RG | QuebecTrees (Q) $mAP_{50:95}$ | $mAR_{50:95}$ | $RF1_{75}$ | OD | RG | OAM-TCD (O) $mAP_{50:95}$ | $mAR_{50:95}$ | $RF1_{75}$ | OD | RG |
|---|---|---|---|---|---|---|---|---|---|---|---|---|---|---|---|---|
| Detectree2-resize | D | 4.09 | 15.67 | N/A | ✓ | -78% | 7.62 | 13.85 | 13.98 | ✓ | -11% | 2.45 | 12.43 | N/A | ✓ | -61% |
| Detectree2-flexi | D+u | 1.75 | 9.86 | N/A | ✓ | -91% | 9.75 | 16.59 | 15.60 | ✓ | 0% | 5.20 | 13.21 | N/A | ✓ | -16% |
| DeepForest | N | 18.06 | 25.82 | N/A | ✗ | 0% | 3.58 | 7.32 | 4.82 | ✓ | -70% | 6.19 | 11.42 | N/A | ✓ | 0% |
| F. R-CNN-RN50 | N | 17.08(±0.31) | 27.16(±0.09) | N/A | ✗ | -6% | 5.97(±0.45) | 18.39(±0.64) | 10.66(±0.19) | ✓ | -32% | 9.75(±0.23) | 18.85(±0.82) | N/A | ✓ | +57% |
| DINO-Swin-L | N | 23.68(±0.20) | 35.18(±0.20) | N/A | ✗ | +31% | 10.46(±2.60) | 23.47(±3.34) | 14.20(±4.13) | ✓ | -9% | 18.42(±1.66) | 29.91(±1.40) | N/A | ✓ | +197% |
| DeepForest | S | 1.16(±0.14) | 5.52(±0.94) | N/A | ✓ | -94% | 21.46(±0.47) | 36.29(±0.25) | 31.09(±1.03) | ✓ | +99% | 9.68(±1.12) | 21.95(±1.25) | N/A | ✓ | +56% |
| F. R-CNN-RN50 | S | 0.63(±0.19) | 2.98(±0.53) | N/A | ✓ | -97% | 17.65(±0.27) | 30.71(±0.46) | 26.10(±0.88) | ✓ | +67% | 8.50(±0.50) | 16.17(±1.09) | N/A | ✓ | +37% |
| DINO-Swin-L | S | 5.16(±0.57) | 14.67(±1.47) | N/A | ✓ | -72% | 27.34(±2.63) | 44.04(±2.69) | 38.34(±2.43) | ✓ | +145% | 22.58(±0.31) | 35.59(±0.52) | N/A | ✓ | +264% |
| DeepForest | NQO | 20.50(±0.26) | 31.13(±0.15) | N/A | ✗ | +13% | 36.75(±0.37) | 49.66(±0.58) | 47.37(±0.22) | ✗ | +203% | 39.00(±0.21) | 49.78(±0.18) | N/A | ✗ | +530% |
| F. R-CNN-RN50 | NQO | 17.94(±0.10) | 28.04(±0.16) | N/A | ✗ | -1% | 33.45(±0.84) | 45.68(±1.02) | 43.65(±0.92) | ✗ | +179% | 38.34(±0.26) | 47.76(±0.31) | N/A | ✗ | +519% |
| DINO-Swin-L | NQO | 23.50(±0.78) | 34.85(±0.80) | N/A | ✗ | +30% | 44.53(±1.19) | 58.48(±1.00) | **56.53(±0.64)** | ✗ | **+262%** | **44.29(±0.33)** | **55.57(±0.41)** | N/A | ✗ | +615% |
| DeepForest | NQOS | 20.71(±0.25) | 32.14(±0.13) | N/A | ✗ | +14% | 36.53(±0.35) | 49.66(±0.55) | 47.04(±0.61) | ✗ | +201% | 38.37(±0.46) | 49.38(±0.24) | N/A | ✗ | +519% |
| F. R-CNN-RN50 | NQOS | 18.47(±0.16) | 28.80(±0.23) | N/A | ✗ | +2% | 31.98(±0.45) | 45.10(±0.31) | 42.06(±0.73) | ✗ | +169% | 38.08(±0.31) | 47.87(±0.28) | N/A | ✗ | +515% |
| DINO-Swin-L | NQOS | **23.90(±0.49)** | **35.53(±0.50)** | N/A | ✗ | +32% | **45.05(±0.59)** | **58.74(±0.56)** | 56.41(±0.87) | ✗ | +261% | 44.03(±0.53) | 55.34(±0.67) | N/A | ✗ | +611% |

**State-of-the-art performance on both tropical and non-tropical datasets.** We present results on temperate and urban forests in Table 5. We observe that both our multi-dataset methods (with and without SELVABOX) outperforms all the other in-distribution or OOD methods on temperate (NeonTreeEvaluation and QuebecTrees) and urban (OAM-TCD) datasets.

Furthermore, training on SELVABOX alone allows our methods to outperform competing approaches on both the QuebecTrees and OAM-TCD datasets, achieving better results on QuebecTrees than models trained on NeonTree (a global-scale temperate forest dataset), demonstrating SELVABOX's quality and the generalization capacity of our training process. We include corresponding qualitative results in Appendix F.6. Our multi-dataset methods reached average performance within standard deviation for non-tropical datasets, confirming that our multi-dataset approach with SELVABOX reaches state-of-the-art performance on both tropical and non-tropical datasets.

## 5.3 ABLATION OF RF1 VS IOU

To better understand how the RF1 metric varies across IoU thresholds other than 0.75, we plotted RF1 as a function of the IoU threshold (0.50 to 0.95) for SELVABOX (Fig. 4) as well as BCI50ha, Detectree2 and QuebecTrees (App. E). NMS hyperparameters were optimized independently on the validation set independently for each IoU threshold. Results on SELVABOX are consistent with Tables 4 and 5: we observe a consistent, substantial gap between our in-distribution DINO-Swin-L variants and competing methods (DeepForest and both Detectree2 methods) across all IoU thresholds. On BCI50ha and Detectree2 datasets, the Detectree2-resize baseline exhibits a local performance peak around IoU = 0.70, sometimes unexpectedly exceeding its scores at lower thresholds (0.50–0.65), which we attribute to

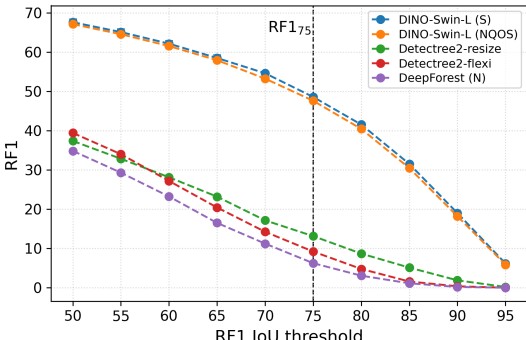

Figure 4: **RF1 vs IoU threshold on SELVABOX.** Comparison of two of our DINO-Swin-L variants and competing methods at different IoU thresholds. In this work we focus on $RF1_{75}$ (IoU 75). For each IoU threshold, NMS hyperparameters are independently optimized on the validation set. Results for other datasets are in Appendix E.

per-threshold NMS tuning and the higher variance induced by the substantially smaller size of these datasets. This behavior further underlines the benefits of SelvaBox's scale, where RF1 is less sensitive to annotation noise and crown size distribution. As a natural extension, we leave for future work the design of an $RF1_{50:95}$ metric, analogous to $mAP_{50:95}$, in which NMS hyperparameters would be tuned against the average RF1 over multiple IoU thresholds.

## 5.4 PRACTICAL ADVICE

We recommend DINO-Swin-L (NQOS) as the default model for most applications and forest types, as it ranks first or a close second on all datasets. The exception is high-resolution tropical drone imagery without water or human constructions, where DINO-Swin-L (S) is preferable. We also recommend tuning NMS hyperparameters, tile extent and ground resolution on a validation set (if available), especially if the trees of interest are either small or very large.

## 6 ETHICAL CONSIDERATIONS AND RESPONSIBLE USE

SELVABOX and the released models are intended to support ecological research and operational forest monitoring (e.g., biodiversity assessment, biomass and carbon-stock estimation), not to facilitate activities such as illegal logging, land grabbing, or other forms of environmentally harmful exploitation, nor actions that could undermine the rights and livelihoods of local and Indigenous communities. We release SELVABOX under a CC-BY 4.0 license and code and model checkpoints under an Apache-2.0 license, both permissive licenses. Although annotations were produced and reviewed by expert biologists, the dataset and resulting models inevitably contain noise and biases, and metrics such as $RF1_{75}$ remain sensitive to annotation completeness and evaluation settings. Outputs from SELVABOX-trained models should therefore be treated as decision-support tools rather than definitive measurements, and not used in isolation for high-stakes management or policy decisions.

## 7 CONCLUSION

We present SELVABOX, the largest tropical tree crown detection dataset to date, with over $83,000$ expert-verified annotations from high-resolution UAV imagery across Central and South American forests. We achieve state-of-the-art performance across in-distribution and out-of-distribution benchmarks in a zero-shot setting training on SELVABOX and other open-access datasets. We advocate for the $RF1_{75}$ metric, a raster-level score reflecting forest monitoring needs, and suggest that future work explore an IoU-averaged $RF1_{50:95}$ metric, as well as alternative aggregation methods such as soft-NMS (Bodla et al., 2017) or weighted boxes fusion (Solovyev et al., 2021). Our dataset, code, and models are fully open to support research in forest monitoring, while acknowledging the potential risks of misuse for illegal exploitation.

## REPRODUCIBILITY STATEMENT

All code, data, and experimental details required to reproduce the results of this paper are made available. The SELVABOX dataset is described in Sections 3 and 4, with additional details on orthomosaics, splits and annotations in App. A. The ML-ready SELVABOX dataset is available on HuggingFace and linked on the first page of this manuscript. The raster-level annotations and AOIs in geopackage format are available on HuggingFace in a separate branch. Preprocessing steps for external datasets benchmarked in this manuscript are described in App. F.1, and we also release these preprocessed versions on HuggingFace. Our open-access data preprocessing package, *geodataset*, and our benchmark, inference and training GitHub repository, *CanopyRS*, are described in App. G and linked on the first page of this manuscript. The main training hyperparameters and compute setup are described in App. B.2. The RF1$_{75}$ metric pseudo-code implementation and related inference hyperparameters used in our benchmarks can be found in App. B.3. Finally, model weights of our best methods as well as smaller model variants are available on HuggingFace and *CanopyRS* package.

## ACKNOWLEDGMENTS

This project was undertaken thanks to funding from IVADO, including the PRF3 project 'AI, biodiversity, and Climate Change', the Canada First Research Excellence Fund, the Canada Research Chair and a Discovery Grant from NSERC to EL, and funding from the Mitacs institute. We thank the many people who helped with the acquisition of data (drone imagery and labels), notably: Sabrina Demers-Thibeault, Vincent Le Falher, Marie-Jeanne Gascon-DeCelles, Simone Aubé, Chloé Fiset, Maxime Têtu-Frégeau, Frédérik Senez, Gonzalo Rivas-Torres, the Outreach Robotics team (especially Hugues Lavigne and Julien Rachiele-Tremblay), Paulo Sérgio, Adriana Simonetti Peixoto, Caroline Vasconcelos, Daniel Magnobosco Marra, Jefferson Hall, Guillaume Tougas, and Isabelle Lefebvre. We also thank Mila for the compute resources.

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

APPENDICES & SUPPLEMENTARY MATERIAL

# A    THE SELVABOX DATASET

## A.1    ORTHOMOSAICS.

The RGB orthomosaics were generated in Agisoft Metashape version 2.1. Images were acquired by flying at a constant elevation above the canopy. We kept a forward overlap of $> 80\%$ and a side overlap of $> 70\%$. Images were acquired around mid-day to minimize shadows. Sky conditions ranged from full sun to overcast.

The main Metashape parameters used for all of our orthomosaic reconstructions were:

- Alignment accuracy: High
- Point cloud quality: High
- Point cloud filtering: Disabled
- Orthomosaic blending mode: Mosaic

Table 6: **SELVABOX orthomosaics**. We denote each type of DJI drone as 'm3e' for Mavic 3 Enterprise, 'm3m' for Mavic 3 Multispectral, 'mavicpro' for Mavic Pro, 'mini2' for Mavic Mini 2.

| Raster name | Drone | Country | Date | Sky conditions | GSD (cm/px) | Forest type | #Hectares | #Annotations | Proposed split(s) |
|---|---|---|---|---|---|---|---|---|---|
| zf2quad | m3m | Brazil | 2024-01-30 | clear | 2.3 | primary | 15.5 | 1343 | valid |
| zf2tower | m3m | Brazil | 2024-01-30 | clear | 2.2 | primary | 9.5 | 1716 | test |
| zf2transectew | m3m | Brazil | 2024-01-30 | clear | 1.5 | primary | 2.6 | 359 | train |
| zf2campinarana | m3m | Brazil | 2024-01-31 | clear | 2.3 | primary | 66 | 16396 | train |
| transectotoni | mavicpro | Ecuador | 2017-08-10 | cloudy | 4.3 | primary | 4.3 | 5119 | train |
| tbslake | m3m | Ecuador | 2023-05-25 | clear | 5.1 | primary | 19 | 1279 | train, test |
| sanitower | mini2 | Ecuador | 2023-09-11 | cloudy | 1.8 | primary | 5.8 | 1721 | train |
| inundated | m3e | Ecuador | 2023-10-18 | cloudy | 2.2 | primary | 68 | 9075 | train, valid, test |
| pantano | m3e | Ecuador | 2023-10-18 | cloudy | 1.9 | primary | 41 | 4193 | train |
| terrafirme | m3e | Ecuador | 2023-10-18 | clear | 2.4 | primary | 110 | 6479 | train |
| asnortheast | m3m | Panama | 2023-12-07 | partial cloud | 1.3 | plantations, secondary | 33 | 12930 | train, valid, test |
| asnorthnorth | m3m | Panama | 2023-12-07 | cloud | 1.2 | plantations, secondary | 15 | 6020 | train |
| asforestnorthe2 | m3m | Panama | 2023-12-08 | clear | 1.5 | secondary | 20 | 5925 | valid, test |
| asforestsouth2 | m3m | Panama | 2023-12-08 | clear | 1.6 | secondary | 28 | 10582 | train |

Table 7: **SELVABOX boxes details.** Details of number of boxes for each raster, country and overall as well as their minimum, maximum and median box size expressed in meters.

| Country | Location | Raster name | # Boxes | Min box size (m) | Max box size (m) | Median box size (m) |
|---|---|---|---|---|---|---|
| Brazil | ZF2 | 20240130_zf2quad_m3m | 1343 | 1.02 | 33.00 | 6.34 |
| | | 20240130_zf2tower_m3m | 1716 | 0.97 | 28.71 | 6.16 |
| | | 20240130_zf2transectew_m3m | 359 | 0.90 | 26.94 | 5.12 |
| | | 20240131_zf2campirana_m3m | 16396 | 0.93 | 36.72 | 6.01 |
| | | All rasters | 19814 | 0.90 | 36.72 | 6.03 |
| Ecuador | Agua Salud | 20231018_inundated_m3e | 9075 | 0.52 | 54.27 | 6.41 |
| | | 20231018_pantano_m3e | 4193 | 0.92 | 41.60 | 6.66 |
| | | 20231018_terrafirme_m3e | 6479 | 0.81 | 53.19 | 6.26 |
| | | 20170810_transectotoni_mavicpro | 5119 | 0.83 | 47.97 | 5.80 |
| | | 20230525_tbslake_m3e | 1279 | 1.46 | 41.28 | 8.45 |
| | | 20230911_sanitower_mini2 | 1721 | 0.86 | 57.16 | 5.53 |
| | | All rasters | 27866 | 0.52 | 57.16 | 6.31 |
| Panama | Agua Salud | 20231208_asforestnorthe2_m3m | 5925 | 0.51 | 36.17 | 4.99 |
| | | 20231207_asnortheast_amsunclouds_m3m | 12930 | 0.50 | 36.42 | 4.17 |
| | | 20231207_asnorthnorth_pmclouds_m3m | 6020 | 0.50 | 29.28 | 4.63 |
| | | 20231208_asforestsouth2_m3m | 10582 | 0.83 | 38.92 | 4.83 |
| | | All rasters | 35457 | 0.50 | 38.92 | 4.58 |
| All | All | All rasters | 83137 | 0.50 | 57.16 | 5.44 |

## A.2 Spatially Separated Splits.

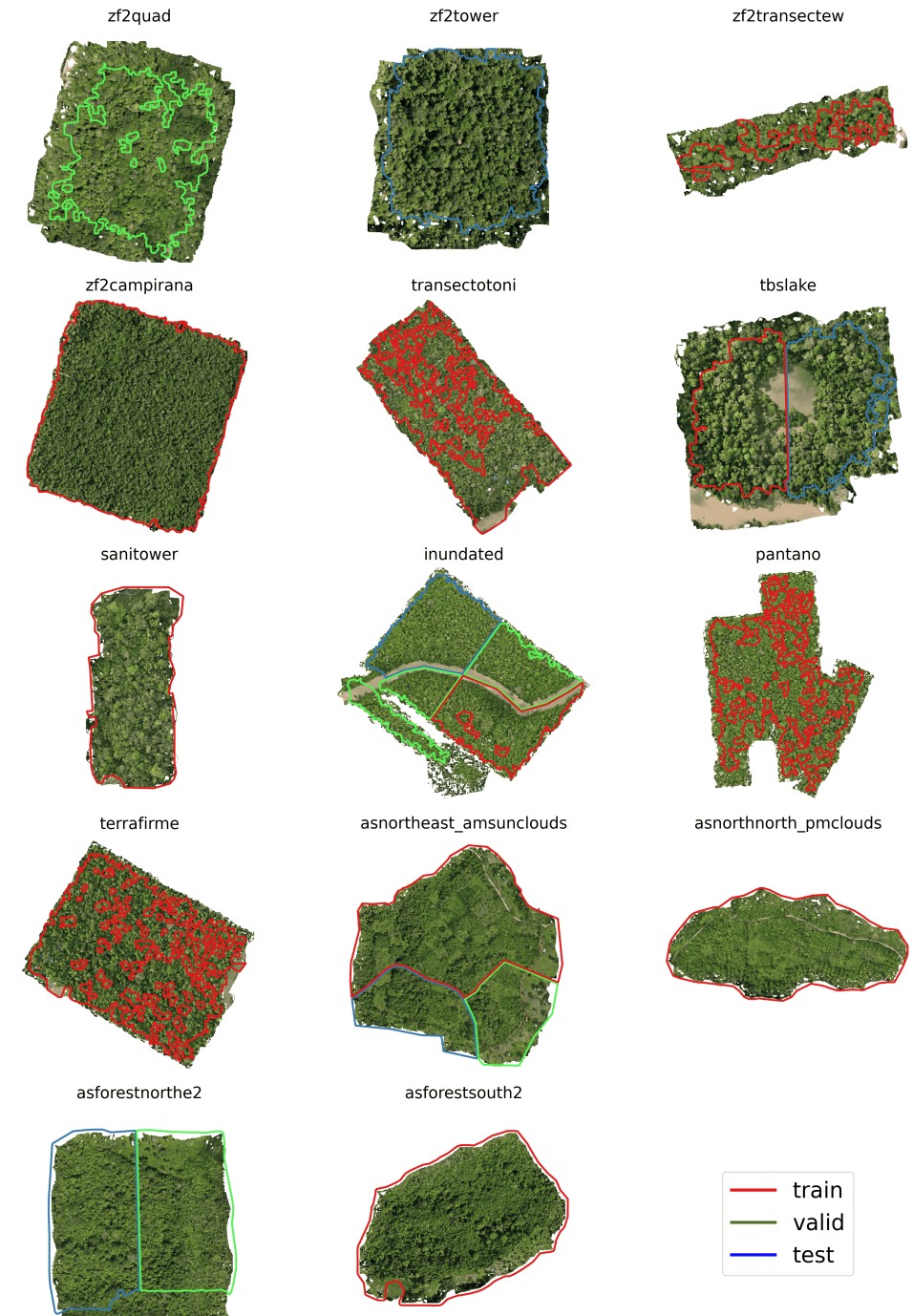

Figure 5: **Visualization of spatially separated splits.** All 14 rasters of SELVABOX are illustrated with their corresponding train, valid and test AOI-based splits. Images are uniformly sized and not at scale. A few train AOIs (red) have holes to exclude sparse annotations (see Section 3).

A.3    ANNOTATION PROTOCOL

The annotations were created by five domain experts with exact same instructions, and all started with a demo and an annotation practice beforehand. All annotations were made in ArcGIS Pro version 3.0 with ArcGIS Online layers to track the online work of two annotators working on the same orthomosaic simultaneously. In large and dense areas, one or several annotators performed an additional pass over the orthomosaic to annotate potential missing trees.

Once annotations were completed by one or several annotators, one or two domain experts performed quality control steps for all annotations of each orthomosaic by following precise guidelines:

1. Set up a 60×60 m grid over the orthomosaic.
2. Proceed to the verification by systematically scanning each cell to avoid missing any areas.
3. Ensure that there are as many annotated trees as possible in each cell.
4. Also annotate dead or leafless trees.
5. Check that annotations already completed are correct, adjusting them if necessary.

All annotators and reviewers were provided with documentation covering difficult use cases as a reference when they were uncertain about the annotation procedure. For example, they were asked to discuss difficult cases with each other and reach consensus, particularly for ambiguous situations such as intertwined crowns, branches adjacent to large crowns that may correspond to separate understory trees, or vegetation that could be lianas rather than individual trees. Some variability in bounding box tightness (slightly more or less padding around crowns) is also expected. However, none of these concerns were flagged as significant during our systematic quality control.

As a comparison, we point out that annotations in OAM-TCD (Veitch-Michaelis et al., 2024) (NeurIPS 2024) were created by professional annotators who were not domain experts, and only a portion of these annotations were reviewed by ecologists.

A.4    INCOMPLETE ANNOTATIONS.

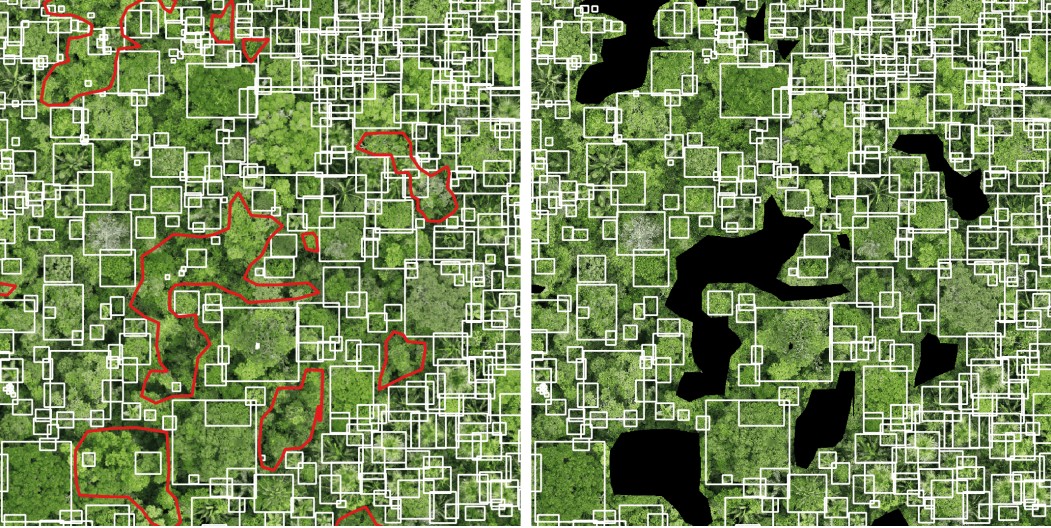

Figure 6: **Example of masked pixels in sparse annotations zones.** Example on a $3555 \times 3555$ pixels training tile ($160 \times 160$ meters) from the *pantano* raster. On the left is the raw tile, showing holes (red polygons) in the train AOI geopackage where annotations (white boxes) are sparse. On the right is the preprocessed tile, where pixels overlapping the AOI holes have been masked to remove sparse annotations. AOI holes were created mostly where visible trees were not annotated (see Section 3).

# B  HYPERPARAMETERS AND AUGMENTATIONS

## B.1  AUGMENTATIONS

For all experiments, we use the same set of basic augmentations:

Table 8: **Settings of data augmentations used for all experiments.** Augmentations were applied in the top to bottom order of the table. The Hue augmentation is applied to pixel values in the 0–255 range. The fallback value column describes the behavior of the preprocessing pipeline when an augmentation is not applied. Multi-dataset models use the multi-res. variants of crop and resize augmentations. The 'spatial-extent' for our single-res. experiments on SELVABOX is either 40 m or 80 m (see Tab. 2 and 3). The crop augmentation for the multi-res. settings is expressed in pixels, where the value is randomly drawn between $x_{\min}$ and $x_{\max}$ that will correspond to different spatial extents depending on the dataset (see Fig. 7 and Sec. F.1). The resize augmentation will either be applied with a fixed value $y$, expressed in pixel, for the single-res. applications on SELVABOX, or randomly drawn between $y_{\min}$ and $y_{\max}$ for the multi-resolution and multi-dataset training approaches.

| Augmentation | Probability | Augmentation Range | Fallback value |
|---|---|---|---|
| Flip Horizontal | 0.5 | — | — |
| Flip Vertical | 0.5 | — | — |
| Rotation | 0.5 | $[-30°, +30°]$ | — |
| Brightness | 0.5 | $[-20\%, +20\%]$ | — |
| Contrast | 0.5 | $[-20\%, +20\%]$ | — |
| Saturation | 0.5 | $[-20\%, +20\%]$ | — |
| Hue | 0.3 | $[-10, +10]$ | — |
| Crop (single-res.) | 0.5 | spatial extent $\times$ $[-10\%, +10\%]$ | spatial extent |
| Crop (multi-res.) | 0.5 | $[x_{\min}, x_{\max}]$ | max. image size |
| Resize (single-res.) | 1.0 | $y$ | — |
| Resize (multi-res.) | 1.0 | $[y_{\min}, y_{\max}]$ | — |

## B.2 TRAINING HYPERPARAMETERS

This section lists the hyperparameters found for each of our settings. We performed grid search ($\approx 10$ hyperparameter combinations) for every setting on four hyperparameters – the learning-rate, its scheduler, the total number of epochs and the batch size. We left all other hyperparameters at their default values as specified in Detectron2 and Detrex configuration files. CosineLR refers to a cosine learning-rate schedule without restart. We applied a 5 000-step warmup at the start of each training session. Training was performed on either 48 GB NVIDIA RTX 8000 or L40S GPUs, depending on compute-cluster availability. Most sessions used one or two GPUs; however, DINO + Swin L-384 with large input sizes, multi-resolution, or multi-dataset settings required four GPUs (one image per GPU per batch) due to their high memory footprint.

Table 9: **Hyperparameters selected for the input size and GSD experimental analyses on SELVABOX.** Hyperparameters selected for each method and spatial extent in Tables 2 and 3. An initial search shown that, for each architecture and spatial extent, the optimal hyperparameters were nearly identical across GSDs; accordingly, we applied the same settings to all GSDs within each spatial extent.

| Method | Extent (m) | Optimizer | LR | Scheduler | Max Epochs | Batch Size |
|---|---|---|---|---|---|---|
| Faster R-CNN (ResNet50) | $40 \times 40$ | SGD | $5 \times 10^{-3}$ | CosineLR | 500 | 8 |
| DINO 4-scale (ResNet50) | $40 \times 40$ | AdamW | $1 \times 10^{-4}$ | CosineLR | 200 | 4 |
| DINO 5-scale (Swin L-384) | $40 \times 40$ | AdamW | $5 \times 10^{-5}$ | CosineLR | 500 | 8 |
| Faster R-CNN (ResNet50) | $80 \times 80$ | SGD | $5 \times 10^{-3}$ | CosineLR | 500 | 4 |
| DINO 4-scale (ResNet50) | $80 \times 80$ | AdamW | $1 \times 10^{-4}$ | CosineLR | 500 | 4 |
| DINO 5-scale (Swin L-384) | $80 \times 80$ | AdamW | $1 \times 10^{-4}$ | CosineLR | 500 | 4 |

Table 10: **Hyperparameters selected for the multi-resolution experimental analysis on SELVABOX.** These hyperparameters were optimal as being the same ones as used for DINO 5-scale (Swin L-384) at $80 \times 80$ m spatial extent. The associated models performance are in Figures 3, 8 and Table 18.

| Method | Train Crop Range (m) | Optimizer | LR | Scheduler | Max Epochs | Batch Size |
|---|---|---|---|---|---|---|
| DINO 5-scale (Swin L-384) | $[36, 88]$ | AdamW | $1 \times 10^{-4}$ | CosineLR | 500 | 4 |
| DINO 5-scale (Swin L-384) | $[30, 100]$ | AdamW | $1 \times 10^{-4}$ | CosineLR | 500 | 4 |
| DINO 5-scale (Swin L-384) | $[30, 120]$ | AdamW | $1 \times 10^{-4}$ | CosineLR | 500 | 4 |

Table 11: **Hyperparameters selected for the OOD experimental analyses with multi-dataset trainings.** For the MultiStepLR scheduler, we reduced the learning rate by a factor of 10 at $80\%$ and again at $90\%$ of the total training epochs. The associated models performance are in Tables 4 and 5.

| Method | Train Datasets | Optimizer | LR | Scheduler | Max Epochs | Batch Size |
|---|---|---|---|---|---|---|
| DeepForest | N | N/A | N/A | N/A | N/A | N/A |
| Faster R-CNN (ResNet50) | N | SGD | $5 \times 10^{-3}$ | CosineLR | 500 | 8 |
| DINO 5-scale (Swin L-384) | N | AdamW | $1 \times 10^{-4}$ | CosineLR | 80 | 4 |
| DeepForest | S | SGD | $5 \times 10^{-3}$ | CosineLR | 500 | 8 |
| Faster R-CNN (ResNet50) | S | SGD | $5 \times 10^{-3}$ | CosineLR | 500 | 8 |
| DINO 5-scale (Swin L-384) | S | AdamW | $1 \times 10^{-4}$ | CosineLR | 500 | 4 |
| DeepForest | N+Q+O | SGD | $2 \times 10^{-3}$ | CosineLR | 200 | 4 |
| Faster R-CNN (ResNet50) | N+Q+O | SGD | $5 \times 10^{-3}$ | CosineLR | 200 | 4 |
| DINO 5-scale (Swin L-384) | N+Q+O | AdamW | $1 \times 10^{-4}$ | CosineLR | 80 | 4 |
| DeepForest | N+Q+O+S | SGD | $5 \times 10^{-3}$ | CosineLR | 120 | 8 |
| Faster R-CNN (ResNet50) | N+Q+O+S | SGD | $5 \times 10^{-3}$ | CosineLR | 120 | 8 |
| DINO 5-scale (Swin L-384) | N+Q+O+S | AdamW | $1 \times 10^{-4}$ | MultiStepLR | 80 | 4 |

### B.3 INFERENCE HYPERPARAMETERS

We detail the pseudocode for the $RF1_{75}$ metric in Algorithm 1 (see Section 4). Setting $\tau_{iou} = 0.75$ corresponds to $RF1_{75}$. Before applying the NMS, we discard predictions whose bounding box lies within a 5%–wide band along the tiles borders. We perform a grid search on the valid set over the non-maximum suppression IoU threshold $\tau_{nms}$ and the minimum detection confidence score $s_{min}$, each taking values in the discrete set $\{0.00, 0.05, 0.10, \ldots, 1.00\}$. We multiprocess the grid search on 12 CPU cores to speed up the process. After finding the optimal $\tau_{nms}$ and $s_{min}$ on the best model seed, we apply it on the test set to all model seeds to compute the final $RF1_{75}$ score with standard deviation.

---

**Algorithm 1** Per-dataset evaluation with weighted RF1

---

**Require:** Dataset $\mathcal{D}$ of rasters, detector $\mathcal{M}$, $\tau_{nms}$, $s_{min}$, $\tau_{iou}$

1:   $\mathcal{R} \leftarrow \emptyset$                                            ▷ list of per-raster F1 scores
2:   $\mathcal{W} \leftarrow \emptyset$                                        ▷ list of per-raster truth counts
3:   **for** each raster $r \in \mathcal{D}$ **do**
4:       $P \leftarrow \emptyset$                                      ▷ accumulate tile preds
5:       $G \leftarrow \mathrm{LoadGroundTruth}(r)$              ▷ load geo-truth
6:       **for** each tile $t$ in $r$ **do**
7:          $p \leftarrow \mathcal{M}.\mathrm{predict}(t)$
8:          $P \leftarrow P \cup p$
9:       **end for**
10:     $P_{conf} \leftarrow \{p \in P : p.\mathrm{score} \geq s_{min}\}$
11:     $P' \leftarrow \mathrm{NonMaxSuppression}(P_{conf}, \tau_{nms})$
12:     $(tp, fp, fn) \leftarrow \mathrm{GreedyMatch}(P', G, \tau_{iou})$
13:     $\mathrm{precision} \leftarrow tp/(tp + fp)$
14:     $\mathrm{recall} \leftarrow tp/(tp + fn)$
15:     $\mathrm{f1} \leftarrow 2 \frac{\mathrm{precision} \, \mathrm{recall}}{\mathrm{precision} + \mathrm{recall}}$
16:     $n \leftarrow |G|$                                      ▷ truth count
17:     $\mathcal{R} \leftarrow \mathcal{R} \cup \mathrm{f1}$
18:     $\mathcal{W} \leftarrow \mathcal{W} \cup n$
19: **end for**
20: $W \leftarrow \sum_{n \in \mathcal{W}} n$
21: $\mathrm{RF1} \leftarrow \frac{1}{W} \sum_{i=1}^{|\mathcal{R}|} \mathcal{R}_i \cdot \mathcal{W}_i$
22: **store** weighted-average RF1

---

---

**Algorithm 2** Greedy matching for RF1

---

1:   **procedure** GREEDYMATCH$(P', G, \tau_{iou})$
2:       sort $P'$ by descending score
3:       mark all $g \in G$ as `unmatched`
4:       $tp \leftarrow 0, \quad fp \leftarrow 0$
5:       **for** each prediction $p \in P'$ **do**
6:          $g^* \leftarrow \arg\max_{g \in G \, : \, g.\mathrm{unmatched}=\mathrm{true}} \mathrm{IoU}(p, g)$
7:          **if** $\mathrm{IoU}(p, g^*) \geq \tau_{iou}$ **then**
8:             $tp \leftarrow tp + 1$
9:             mark $g^*$ as `matched`
10:         **else**
11:            $fp \leftarrow fp + 1$
12:         **end if**
13:       **end for**
14:       $fn \leftarrow \big|\{g \in G : g.\mathrm{unmatched} = \mathrm{true}\}\big|$
15:       **return** $(tp, fp, fn)$
16: **end procedure**

---

Table 12: **Optimal inference hyperparameters for the input size and GSD experimental analysis at** $40\times40$ **meters on SELVABOX.** Both optimal NMS and score thresholds are selected by maximizing the RF1$_{75}$ metric as described in Algorithm 1. The associated models performance are in Table 2.

| Method | GSD | I. size | NMS IoU ($\tau_{nms}$) | Score thr. ($s_{min}$) |
|---|---|---|---|---|
| Faster RCNN ResNet50 | 10 | 400 | 0.50 | 0.85 |
| | 10 | 666 | 0.60 | 0.70 |
| | 10 | 888 | 0.50 | 0.80 |
| | 6 | 666 | 0.55 | 0.90 |
| | 6 | 888 | 0.70 | 0.90 |
| | 4.5 | 888 | 0.65 | 0.85 |
| DINO 4-scale ResNet50 | 10 | 400 | 0.70 | 0.45 |
| | 10 | 666 | 0.50 | 0.35 |
| | 10 | 888 | 0.75 | 0.35 |
| | 6 | 666 | 0.65 | 0.45 |
| | 6 | 888 | 0.35 | 0.35 |
| | 4.5 | 888 | 0.65 | 0.40 |
| DINO 5-scale Swin L-384 | 10 | 400 | 0.75 | 0.35 |
| | 10 | 666 | 0.80 | 0.45 |
| | 10 | 888 | 0.35 | 0.35 |
| | 6 | 666 | 0.55 | 0.35 |
| | 6 | 888 | 0.45 | 0.40 |
| | 4.5 | 888 | 0.50 | 0.35 |

Table 13: **Optimal inference hyperparameters for the input size and GSD experimental analysis at** $80\times80$ **meters on SELVABOX.** Both optimal NMS and score thresholds are selected by maximizing the RF1$_{75}$ metric on the validation set of SELVABOX as described in Algorithm 1. The associated models performance are in Table 3.

| Method | GSD | I. size | NMS IoU ($\tau_{nms}$) | Score thr. ($s_{min}$) |
|---|---|---|---|---|
| Faster RCNN ResNet50 | 10 | 800 | 0.70 | 0.75 |
| | 10 | 1333 | 0.40 | 0.70 |
| | 10 | 1777 | 0.35 | 0.60 |
| | 6 | 1333 | 0.40 | 0.70 |
| | 6 | 1777 | 0.45 | 0.75 |
| | 4.5 | 1777 | 0.25 | 0.35 |
| DINO 4-scale ResNet50 | 10 | 800 | 0.35 | 0.45 |
| | 10 | 1333 | 0.75 | 0.45 |
| | 10 | 1777 | 0.70 | 0.40 |
| | 6 | 1333 | 0.35 | 0.40 |
| | 6 | 1777 | 0.75 | 0.35 |
| | 4.5 | 1777 | 0.40 | 0.35 |
| DINO 5-scale Swin L-384 | 10 | 800 | 0.75 | 0.35 |
| | 10 | 1333 | 0.80 | 0.40 |
| | 10 | 1777 | 0.70 | 0.35 |
| | 6 | 1333 | 0.75 | 0.45 |
| | 6 | 1777 | 0.65 | 0.35 |
| | 4.5 | 1777 | 0.75 | 0.45 |

Table 14: **Optimal inference hyperparameters for the multi-resolution experimental analysis on** **SELVABOX.** Both optimal NMS and score thresholds are selected by maximizing the RF1$_{75}$ metric on the validation set of SELVABOX as described in Algorithm 1. The associated models performance are in Figures 3, 8 and Table 18.

| Method | Train Crop Range (m) | Test GSD (cm) | NMS IoU ($\tau_{\mathrm{nms}}$) | Score thr. ($s_{\min}$) |
|---|---|---|---|---|
| DINO 5-scale Swin L-384 | [36, 88] | 10 | 0.70 | 0.45 |
| | | 6 | 0.60 | 0.45 |
| | | 4.5 | 0.70 | 0.45 |
| DINO 5-scale Swin L-384 | [30, 100] | 10 | 0.70 | 0.40 |
| | | 6 | 0.70 | 0.40 |
| | | 4.5 | 0.60 | 0.40 |
| DINO 5-scale Swin L-384 | [30, 120] | 10 | 0.70 | 0.40 |
| | | 6 | 0.50 | 0.35 |
| | | 4.5 | 0.80 | 0.40 |

Table 15: **Optimal inference hyperparameters for the experimental analyses with multi-dataset** **trainings.** Both optimal NMS and score thresholds are selected by maximizing the RF1$_{75}$ metric on the validation sets of both SELVABOX and Detectree2 as described in Algorithm 1. The associated models performance are in Tables 4 and 5.

| Method | Train dataset(s) | NMS IoU ($\tau_{\mathrm{nms}}$) | Score thr. ($s_{\min}$) |
|---|---|---|---|
| Detectree2-resize | D | 0.30 | 0.25 |
| Detectree2-flexi | D+urban | 0.80 | 0.20 |
| DeepForest | N | 0.80 | 0.05 |
| F. R-CNN-ResNet50 | N | 0.10 | 0.50 |
| DINO-Swin-L | N | 0.80 | 0.55 |
| DeepForest | S | 0.30 | 0.40 |
| F. R-CNN-ResNet50 | S | 0.20 | 0.45 |
| DINO-Swin-L | S | 0.80 | 0.40 |
| DeepForest | N+Q+O | 0.70 | 0.40 |
| F. R-CNN-ResNet50 | N+Q+O | 0.50 | 0.45 |
| DINO-Swin-L | N+Q+O | 0.70 | 0.40 |
| DeepForest | N+Q+O+S | 0.30 | 0.40 |
| F. R-CNN-ResNet50 | N+Q+O+S | 0.20 | 0.50 |
| DINO-Swin-L | N+Q+O+S | 0.70 | 0.50 |

## C  BENCHMARKING RESOLUTIONS AND IMAGE SIZES

Table 16: **Model, resolution and spatial extent selection on SELVABOX at** $40 \times 40$ **m.** Comparison of performances on the proposed test set of SELVABOX with variable tile spatial extent. Tile size and ground spatial distance (GSD) are in cm. We highlight results per method and backbone as ▮ the first, ▮ the second and ▮ the third best scores. We also **bold** and underline the best and second best scores overall. Note that $mAP_{50}$, $mAP_{50:95}$, $mAR_{50}$ and $mAR_{50:95}$ cannot be compared between $40 \times 40$ m and $80 \times 80$ m inputs as images do not match, but we can use $RF1_{75}$ to compare final post-aggregation results at the raster-level.

| Method | GSD | I. size | $mAP_{50}$ | $mAP_{50:95}$ | $mAR_{50}$ | $mAR_{50:95}$ | $RF1_{75}$ |
|---|---|---|---|---|---|---|---|
| Faster RCNN ResNet50 | 10 | 400 | 54.92 (±0.08) | 26.90 (±0.13) | 74.48 (±0.42) | 40.87 (±0.35) | 35.78 (±0.44) |
| | 10 | 666 | 57.03 (±0.08) | 28.40 (±0.13) | 76.53 (±0.49) | 42.79 (±0.19) | 37.75 (±0.30) |
| | 10 | 888 | 56.42 (±0.30) | 28.51 (±0.20) | 76.21 (±0.14) | 43.36 (±0.19) | 37.46 (±0.91) |
| | 6 | 666 | 57.13 (±0.17) | 29.31 (±0.05) | 76.25 (±0.66) | 43.59 (±0.20) | 39.97 (±0.33) |
| | 6 | 888 | 57.27 (±0.54) | 29.40 (±0.34) | 77.26 (±0.77) | 44.18 (±0.44) | 38.92 (±0.51) |
| | 4.5 | 888 | 58.33 (±0.21) | 30.25 (±0.24) | 78.41 (±0.15) | 45.18 (±0.30) | 39.97 (±0.67) |
| DINO 4-scale ResNet50 | 10 | 400 | 56.98 (±0.25) | 30.63 (±0.24) | 76.92 (±0.74) | 48.06 (±0.33) | 41.14 (±0.80) |
| | 10 | 666 | 57.62 (±0.64) | 31.76 (±0.86) | 78.56 (±0.16) | 50.40 (±0.55) | 41.57 (±1.94) |
| | 10 | 888 | 58.11 (±0.64) | 32.19 (±0.33) | 78.55 (±0.34) | 50.68 (±0.19) | 42.47 (±0.97) |
| | 6 | 666 | 58.71 (±0.34) | 33.46 (±0.22) | 78.95 (±0.26) | 51.80 (±0.31) | 44.55 (±0.18) |
| | 6 | 888 | 58.78 (±0.51) | 33.54 (±0.40) | 79.16 (±0.02) | 52.12 (±0.18) | 43.34 (±0.79) |
| | 4.5 | 888 | 60.11 (±0.36) | 34.19 (±0.13) | 79.87 (±0.15) | 52.53 (±0.40) | 44.26 (±0.83) |
| DINO 5-scale Swin L-384 | 10 | 400 | 60.44 (±0.32) | 33.84 (±0.20) | 79.84 (±0.29) | 52.02 (±0.25) | 45.37 (±0.23) |
| | 10 | 666 | 61.26 (±0.30) | 34.64 (±0.25) | 80.77 (±0.17) | 52.91 (±0.30) | 46.39 (±0.52) |
| | 10 | 888 | 61.06 (±0.55) | 34.92 (±0.34) | 80.70 (±0.13) | 53.23 (±0.14) | 45.22 (±0.70) |
| | 6 | 666 | 62.91 (±0.46) | 37.07 (±0.16) | 81.58 (±0.12) | 55.18 (±0.22) | 48.50 (±0.60) |
| | 6 | 888 | 62.45 (±0.17) | 36.22 (±0.38) | 81.47 (±0.18) | 54.55 (±0.43) | 48.13 (±0.60) |
| | 4.5 | 888 | **63.41** (±0.29) | **37.78** (±0.15) | **82.33** (±0.35) | **56.30** (±0.21) | **49.76** (±0.43) |

Table 17: **Model, resolution and spatial extent selection on SELVABOX at** $80 \times 80$ **m.** Comparison of performances on the proposed test set of SELVABOX with variable tile spatial extent. Tile size and ground spatial distance (GSD) are in cm. We highlight results per method and backbone as ▮ the first, ▮ the second and ▮ the third best scores. We also **bold** and underline the best and second best scores overall. Note that $mAP_{50}$, $mAP_{50:95}$, $mAR_{50}$ and $mAR_{50:95}$ cannot be compared between $40 \times 40$ m and $80 \times 80$ m inputs as images do not match, but we can use $RF1_{75}$ to compare final post-aggregation results at the raster-level.

| Method | GSD | I. size | $mAP_{50}$ | $mAP_{50:95}$ | $mAR_{50}$ | $mAR_{50:95}$ | $RF1_{75}$ |
|---|---|---|---|---|---|---|---|
| Faster RCNN ResNet50 | 10 | 800 | 50.50 (±0.44) | 24.94 (±0.34) | 64.72 (±1.25) | 35.93 (±0.55) | 34.66 (±0.97) |
| | 10 | 1333 | 51.37 (±0.11) | 26.25 (±0.14) | 67.57 (±0.63) | 38.59 (±0.41) | 36.09 (±0.51) |
| | 10 | 1777 | 54.20 (±0.55) | 27.58 (±0.24) | 70.65 (±1.84) | 40.21 (±0.38) | 35.74 (±1.26) |
| | 6 | 1333 | 51.96 (±0.64) | 26.52 (±0.80) | 69.77 (±1.53) | 39.55 (±0.75) | 36.22 (±1.45) |
| | 6 | 1777 | 54.68 (±0.26) | 27.89 (±0.35) | 72.32 (±1.35) | 41.02 (±0.69) | 35.94 (±0.84) |
| | 4.5 | 1777 | 56.21 (±0.76) | 28.74 (±0.44) | 72.12 (±0.76) | 41.27 (±0.59) | 37.52 (±0.58) |
| DINO 4-scale ResNet50 | 10 | 800 | 58.32 (±0.44) | 30.90 (±0.51) | 76.33 (±0.28) | 47.29 (±0.33) | 41.20 (±0.39) |
| | 10 | 1333 | 59.65 (±0.20) | 32.39 (±0.02) | 77.61 (±0.07) | 49.22 (±0.10) | 43.08 (±0.20) |
| | 10 | 1777 | 59.31 (±1.29) | 32.51 (±0.89) | 77.23 (±0.34) | 49.35 (±0.47) | 42.39 (±1.25) |
| | 6 | 1333 | 59.84 (±0.42) | 33.06 (±0.29) | 77.91 (±0.17) | 49.93 (±0.39) | 42.92 (±0.51) |
| | 6 | 1777 | 60.48 (±0.26) | 33.62 (±0.10) | 78.32 (±0.21) | 50.85 (±0.17) | 44.18 (±0.18) |
| | 4.5 | 1777 | 61.09 (±0.45) | 33.81 (±0.84) | 78.93 (±0.32) | 51.00 (±0.77) | 43.26 (±0.45) |
| DINO 5-scale Swin L-384 | 10 | 800 | 62.02 (±0.08) | 33.90 (±0.09) | 78.89 (±0.22) | 50.29 (±0.38) | 44.64 (±0.20) |
| | 10 | 1333 | 61.73 (±0.72) | 34.22 (±0.34) | 79.03 (±0.87) | 50.76 (±0.57) | 45.64 (±1.03) |
| | 10 | 1777 | 62.86 (±0.78) | 35.30 (±0.26) | 79.94 (±0.68) | 52.12 (±0.62) | 45.37 (±0.08) |
| | 6 | 1333 | **64.91** (±0.30) | 37.12 (±0.38) | 81.01 (±0.09) | 53.56 (±0.48) | 47.81 (±0.40) |
| | 6 | 1777 | 63.34 (±0.58) | 35.77 (±0.84) | 80.59 (±0.16) | 52.91 (±0.56) | 45.88 (±1.97) |
| | 4.5 | 1777 | 64.59 (±1.03) | **37.79** (±0.55) | **81.35** (±0.71) | **54.66** (±0.47) | **49.38** (±0.76) |

# D   MULTI-RESOLUTION APPROACH

## D.1   MULTI-RESOLUTION EXAMPLE

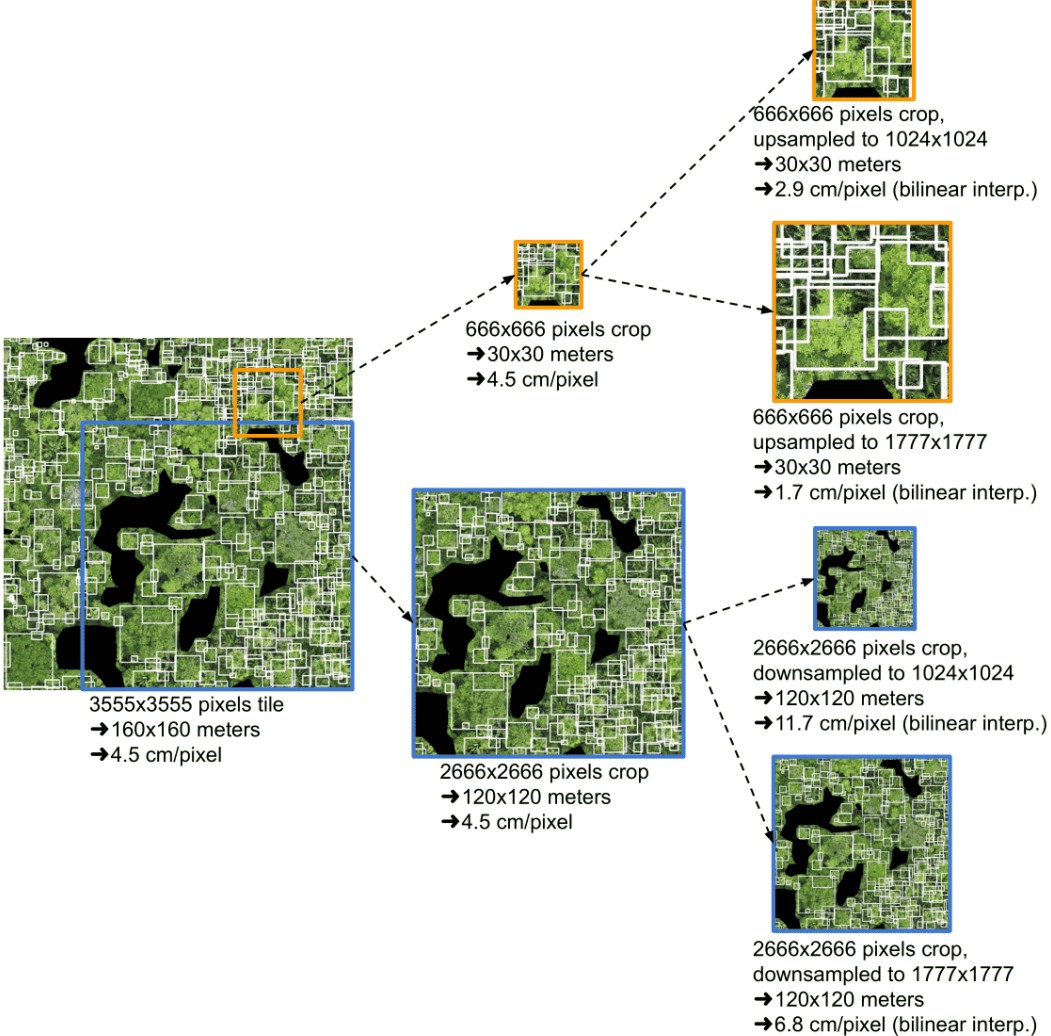

Figure 7: **Example of cropping and resizing augmentations for the multi-resolution approach.** We showcase the $[30, 120]$ m configuration used in our benchmark: a $3555 \times 3555$ tile at $4.5$cm $= 0.045$ m GSD, equivalent to a $160 \times 160$ m spatial extent, will be cropped with a random crop size value in $[666, 2666]$ pixels, and then resized to a random value in $[1024, 1777]$ pixels. This process has two effects: ① cropping performs augmentation for spatial extent – in our example, the original input has the potential to be cropped in a ground extent range of $[30, 120]$ m; ② resizing performs the GSD augmentation – in our example, the largest possible crop (in blue) of 2666 pixels (or 120 m) can be downsampled to $1024 \times 1024$, which yields a maximum effective GSD of $0.045$ m $\times \frac{2666}{1024} = 0.117$ m $= 11.7$ cm per pixel, far from the original 4.5 cm per pixel. Similarly, the smallest possible crop (in orange) of 666 pixels (or 30 m) can be upsampled to $1777 \times 1777$ pixels, yielding a minimum effective GSD of $0.045$ m $\times \frac{666}{1777} = 0.017$ m $= 1.7$ cm per pixel. Note that for small crops, the effective GSD after upsampling (via bilinear interpolation) can fall below the original 4.5 cm/pixel, even though no new image detail is added.

### D.2 MULTI-RESOLUTION ADDITIONAL RESULTS

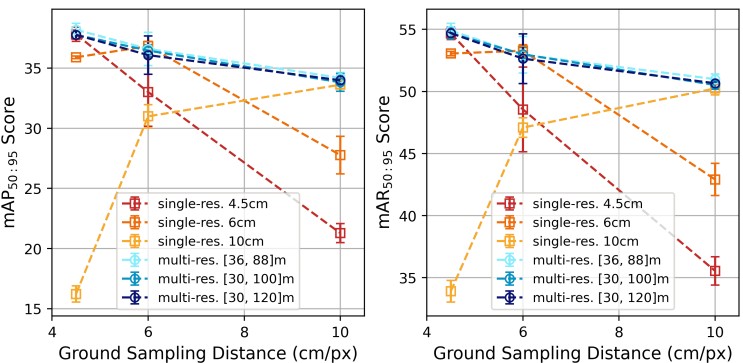

Figure 8: **Multi-resolution vs. single-resolution on SELVABOX.** Comparison of $mAP_{50:95}$ and $mAR_{50:95}$ between best performing single-resolution methods from Table 3 trained with a fixed spatial extent of $80 \times 80$ m, against multi-resolution approaches with increasingly large crop augmentation ranges ([36, 88], [30, 100] and [30, 120]). All methods are 'DINO 5-scale Swin L-384'. It supports results illustrated in Figure 3.

Table 18: **Multi-resolution vs. single-resolution on SELVABOX.** Comparison of best performing methods from Table 3 trained with a fixed spatial extent against multi-resolution approaches. All methods are 'DINO 5-scale Swin L-384', have been trained at 4.5cm. We mark the best and second-best scores in **bold** and underline, respectively. These results are also illustrated in Figures 3 and 8.

| Train extent (m) | Test extent (m) | Test res. (cm/px) | $mAP_{50}$ | $mAP_{50:95}$ | $mAR_{50}$ | $mAR_{50:95}$ | $RF1_{75}$ |
|---|---|---|---|---|---|---|---|
| 80 | 80 | 10 | 62.02 ($\pm$0.08) | 33.90 ($\pm$0.09) | 78.89 ($\pm$0.22) | 50.29 ($\pm$0.38) | 44.64 ($\pm$0.20) |
| 80 | 80 | 6 | 64.91 ($\pm$0.30) | 37.12 ($\pm$0.38) | 81.01 ($\pm$0.09) | 53.56 ($\pm$0.48) | 47.81 ($\pm$0.40) |
| 80 | 80 | 4.5 | 64.59 ($\pm$1.03) | 37.79 ($\pm$0.55) | 81.35 ($\pm$0.71) | 54.66 ($\pm$0.47) | **49.38** ($\pm$0.76) |
| $[36, 88] \cup \{160\}$ | 80 | 10 | 63.33 ($\pm$0.48) | 34.19 ($\pm$0.44) | 79.98 ($\pm$0.21) | 50.99 ($\pm$0.41) | 45.03 ($\pm$0.53) |
| | 80 | 6 | 65.38 ($\pm$0.41) | 36.60 ($\pm$1.38) | 81.29 ($\pm$0.20) | 52.95 ($\pm$1.47) | 47.87 ($\pm$0.92) |
| | 80 | 4.5 | **65.68** ($\pm$0.09) | **38.19** ($\pm$0.54) | 81.85 ($\pm$0.05) | **54.90** ($\pm$0.59) | 49.16 ($\pm$0.06) |
| $[30, 100] \cup \{160\}$ | 80 | 10 | 62.52 ($\pm$1.30) | 33.82 ($\pm$0.74) | 79.42 ($\pm$0.35) | 50.52 ($\pm$0.35) | 44.13 ($\pm$0.73) |
| | 80 | 6 | 64.70 ($\pm$0.48) | 36.46 ($\pm$0.49) | 80.99 ($\pm$0.12) | 52.99 ($\pm$0.55) | 47.96 ($\pm$0.48) |
| | 80 | 4.5 | 65.11 ($\pm$0.28) | 37.77 ($\pm$0.36) | 81.47 ($\pm$0.15) | 54.68 ($\pm$0.47) | 48.79 ($\pm$0.51) |
| $[30, 120] \cup \{160\}$ | 80 | 10 | 62.76 ($\pm$0.49) | 33.99 ($\pm$0.35) | 79.51 ($\pm$0.09) | 50.66 ($\pm$0.08) | 44.91 ($\pm$0.65) |
| | 80 | 6 | 64.44 ($\pm$0.26) | 36.08 ($\pm$1.59) | 80.68 ($\pm$0.42) | 52.64 ($\pm$2.00) | 46.65 ($\pm$1.67) |
| | 80 | 4.5 | 64.92 ($\pm$0.53) | 37.77 ($\pm$0.35) | 81.19 ($\pm$0.08) | 54.69 ($\pm$0.07) | 48.60 ($\pm$0.49) |

# E   ABLATION STUDY ON RF1 IOU THRESHOLD

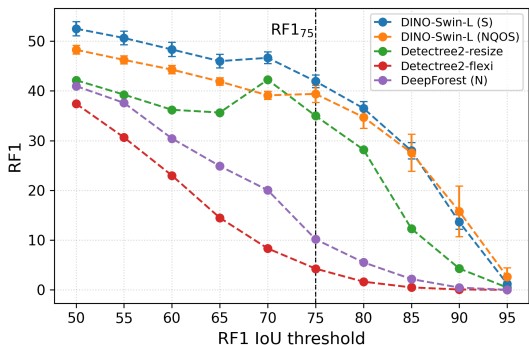

Figure 9: **RF1 vs IoU threshold on BCI50ha.** Comparison of two of our DINO-Swin-L variants and competing methods at different IoU thresholds. In this work we focus on $RF1_{75}$ (IoU=0.75). For each IoU threshold, NMS hyperparameters are re-optimized on the validation set.

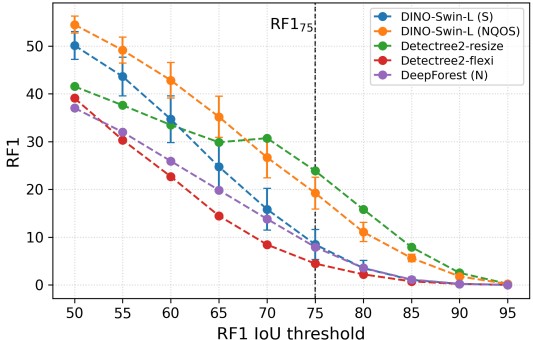

Figure 10: **RF1 vs IoU threshold on Detectree2.** Comparison of two of our DINO-Swin-L variants and competing methods at different IoU thresholds. In this work we focus on $RF1_{75}$ (IoU=0.75). For each IoU threshold, NMS hyperparameters are re-optimized on the validation set.

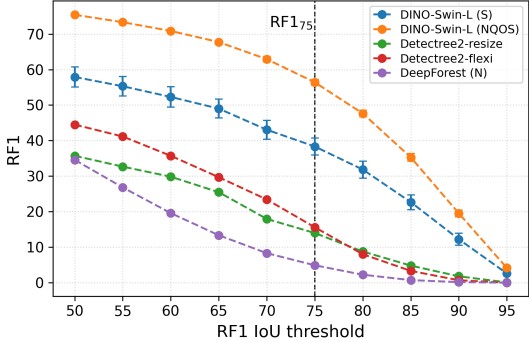

Figure 11: **RF1 vs IoU threshold on QuebecTrees.** Comparison of two of our DINO-Swin-L variants and competing methods at different IoU thresholds. In this work we focus on $RF1_{75}$ (IoU=0.75). For each IoU threshold, NMS hyperparameters are re-optimized on the validation set.

# F   OUT-OF-DISTRIBUTION ANALYSIS

## F.1   EXTERNAL DATASETS PREPROCESSING

For NeonTreeEvaluation, we keep the proposed $400 \times 400$ pixels test inputs at $10\,\mathrm{cm}$ GSD and define train and validation AOIs on their rasters. Similarly, for QuebecTrees, we keep the proposed test split AOI while defining our own train and validation AOIs. As Detectree2's train, validation, and test splits are not shared publicly, we defined our own validation and test AOIs, while keeping the input size as $1000 \times 1000$ to follow their guidelines. BCI50ha is only used for OOD evaluation (see OOD experiments in Sections 4 and 5), so we define test AOIs spanning both rasters.

OAM-TCD contains two types of annotations: individual trees and tree groups. Unfortunately, tree groups would introduce noise during the training process as all other datasets focus on individual tree detection. Therefore, we only consider individual trees annotations and we mask the pixels associated to tree groups from the training data to ensure consistency. This process is similar to how we mask specific low quality pixels and sparse annotations in SELVABOX as detailed in Section 3. OAM-TCD provides five predefined cross-validation folds; we train on folds 0–3 and use fold 4 exclusively for validation. We further divide the $2048 \times 2048$ validation and test tiles of OAM-TCD into $1024 \times 1024$ tiles with 50% overlap, as $204.8 \times 204.8\,\mathrm{m}$ GSD would be significantly larger than other datasets. We refer to Table 19 for more details on final preprocessed datasets statistics and information.

For each dataset divided into tiles, we apply the same AOI-based pixel masking, black/white/transparent pixel cover threshold, and 0-annotation tile removal, as described in Section 3. We use 50% overlap between tiles for all datasets for which we divided rasters into tiles, except BCI50ha where we use 75% to maximize cover for 50+ meters tree crowns (same as SELVABOX test split). We also release these preprocessed external datasets on HuggingFace, including the proposed AOIs and raster-level annotation geopackages for all datasets, in a standardized ML-ready format and with their original CC-BY 4.0 license to ensure reproducibility of our benchmark and facilitate experiments of researchers and practitioners for tree-crown detection. We used version 1.0.0 of OAM-TCD,[4] version v1 of QuebecTrees,[5] version v2 of Detectree2,[6] version 0.2.2 of NeonTreeEvaluation,[7] and version 2 of BCI50ha.[8]

Table 19: **Preprocessing and training parameters for all datasets used.** The SELVABOX parameters correspond to the $[30, 120]\,\mathrm{m}$ multi-resolution setting. Although test tiles outnumber training tiles numerically, training tiles are deliberately larger in spatial extent to facilitate augmentation strategies, resulting in greater total geographic coverage within the train split. The minimum effective train resolution range is reached by using bilinear interpolation from the smallest possible crop size to the largest possible input resize value. *At training time, we resize NeonTreeEvaluation training tiles to 2000 pixels before cropping to ensure that the effective train extent range reaches the $40\,\mathrm{m}$ used in the test split.

| Dataset | GSD (cm/px) | #Train Images | Train size (px) | Augm. Crop range (px) | Augm. Resize range (px) | Effective train extent range (m) | Effective train res. range (cm/px) | #Test Images | Test size (px) | Test extent (m) |
|---|---|---|---|---|---|---|---|---|---|---|
| NeonTreeEvaluation | 10 | 912 | 1200 | [666, 2666] | [1024, 1777] | [40, 120]* | [2.3, 11.7] | 194 | 400 | 40 |
| OAM-TCD | 10 | 3024 | 2048 | [666, 2666] | [1024, 1777] | [66.6, 204.8] | [3.8, 20] | 2527 | 1024 | 102.4 |
| QuebecTrees | 3 | 148 | 3333 | [666, 2666] | [1024, 1777] | $[20, 80] \cup \{100\}$ | [1.1, 9.8] | 168 | 1666 | 50 |
| SELVABOX | 4.5 | 585 | 3555 | [666, 2666] | [1024, 1777] | $[30, 120] \cup \{160\}$ | [1.7, 15.6] | 1477 | 1777 | 80 |
| Detectree2 | 10 | N/A | N/A | N/A | N/A | N/A | N/A | 311 | 1000 | 100 |
| BCI50ha | 4.5 | N/A | N/A | N/A | N/A | N/A | N/A | 2706 | 1777 | 80 |

---

[4]OAM-TCD: https://zenodo.org/records/11617167

[5]QuebecTrees: https://zenodo.org/records/8148479

[6]Detectree2: https://zenodo.org/records/8136161

[7]NeonTreeEvaluation: https://zenodo.org/records/5914554

[8]BCI50ha: Smithsonian Barro Colorado Island 50-ha plot crown maps

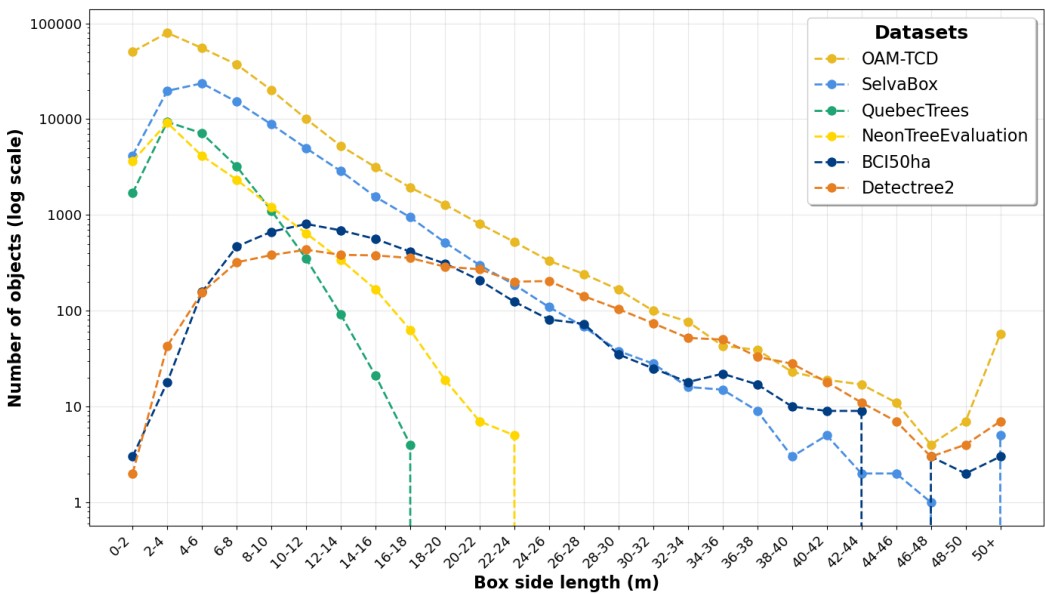

Figure 12: **Distribution of box annotations size across datasets.**

## F.2 STATISTICAL COMPARISON BETWEEN DATASETS

To rigorously compare SelvaBox with existing datasets, we quantified crown-size distribution differences using three complementary statistical approaches:

**Methodology** We computed the Jensen-Shannon (JS) distance and Kullback-Leibler (KL) divergence across datasets. Since KL divergence is asymmetric and unbounded, making interpretation difficult, we prioritize JS distance, which is symmetric, bounded between 0 and 1, and well-suited for discrete distributions. We also performed two-sample Kolmogorov-Smirnov (KS) tests comparing SelvaBox against existing datasets. The KS test evaluates the maximum vertical distance between empirical cumulative distribution functions (ECDFs), providing a non-parametric, distribution-free measure robust to differences in dataset size. Given that KS test p-values follow $p \approx 2e^{-2n \cdot D^2}$ Marsaglia et al. (2003), where $n$ is dataset size and $D$ the KS statistic, and our dataset sizes range from $n = 3,947$ (Detectree2) to $266,663$ (OAM-TCD), we expect p-values approaching $1.04 \cdot 10^{-35}$ when $D \geq 0.1$ and $n \geq 3,947$.

**Results** SelvaBox exhibits substantially different crown-size distributions from tropical datasets (JS Distance: BCI50ha = 0.6248, Detectree2 = 0.6476) and moderately different distributions from temperate datasets (NeonTreeEvaluation = 0.2789, QuebecTrees = 0.2622). Pairwise KS tests reveal highly significant differences across all comparisons (p < 0.001, KS statistics ranging from 0.2117 to 0.6338), confirming that SelvaBox's crown-size distribution is statistically distinct from all existing datasets. The large annotation counts (3,947 to 266,663 samples) ensure these distributional differences are meaningful and robust to dataset scale variations.

**Interpretation** OAM-TCD shows the greatest similarity to SelvaBox (JS Distance = 0.2152), likely because its large geographic scale and multi-biome coverage encompass diverse crown morphologies, unlike datasets restricted to single regions or biomes. The asymmetric KL divergence values further support these conclusions, demonstrating how SelvaBox uniquely captures the crown-size distributions and structural diversity of tropical forests.

Table 20: **Statistical comparison of crown-size distributions between SELVABOX and existing datasets.** Higher JS distance, KL divergence, and KS statistics indicate greater distributional differences, while very small KS $p$-values indicate that the null hypothesis of identical distributions can be rejected.

| | | BCI50ha | Detectree2 | NeonTreeEval. | QuebecTrees | OAM-TCD |
|---|---|---|---|---|---|---|
| SelvaBox | JS Distance | 0.6248 | 0.6476 | 0.2789 | 0.2622 | 0.2152 |
| | KL Divergence | 2.4293 | 2.1779 | 0.3826 | 0.7028 | 0.1687 |
| | KS Test | 0.6231 | 0.6338 | 0.3257 | 0.2270 | 0.2117 |
| | KS Test $p$-value | $< 1 \cdot 10^{-35}$ | $< 1 \cdot 10^{-35}$ | $< 1 \cdot 10^{-35}$ | $< 1 \cdot 10^{-35}$ | $< 1 \cdot 10^{-35}$ |

### F.3 EXTERNAL METHODS EVALUATION

We keep the default Detectree2 inference parameters provided in their python library. For DeepForest, we use their python library directly to benchmark their method but limit input size to $1000 \times 1000$ pixels maximum following their documentation guidelines and examples.

### F.4 REFORESTREE DATASET QUALITATIVE RESULTS.

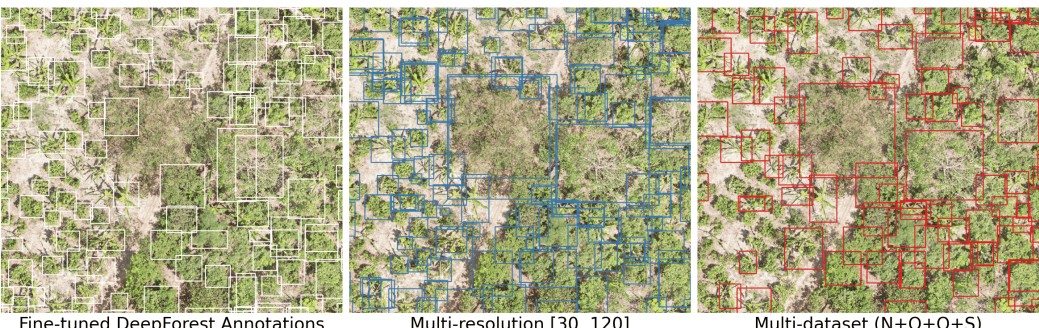

Fine-tuned DeepForest Annotations     Multi-resolution [30, 120]     Multi-dataset (N+Q+O+S)

Figure 13: **Qualitative results on ReforesTree.** In white the ReforesTree annotations generated from an in-distribution and fine-tuned DeepForest model, in blue our best multi-resolution [30, 120] model and in red our best model trained on multi-dataset + SELVABOX (both our methods are OOD). Results are shown post-NMS, using the optimal NMS IoU ($\tau_{\mathrm{nms}}$) and score ($s_{\mathrm{min}}$) thresholds for RF1$_{75}$ from Algorithm 1 (see Section B.3 for exact values). These examples illustrate the superior detection performance of our DINO-Swin models compared to ReforesTree annotations, especially for larger trees.

## F.5 TROPICAL DATASETS QUALITATIVE RESULTS.

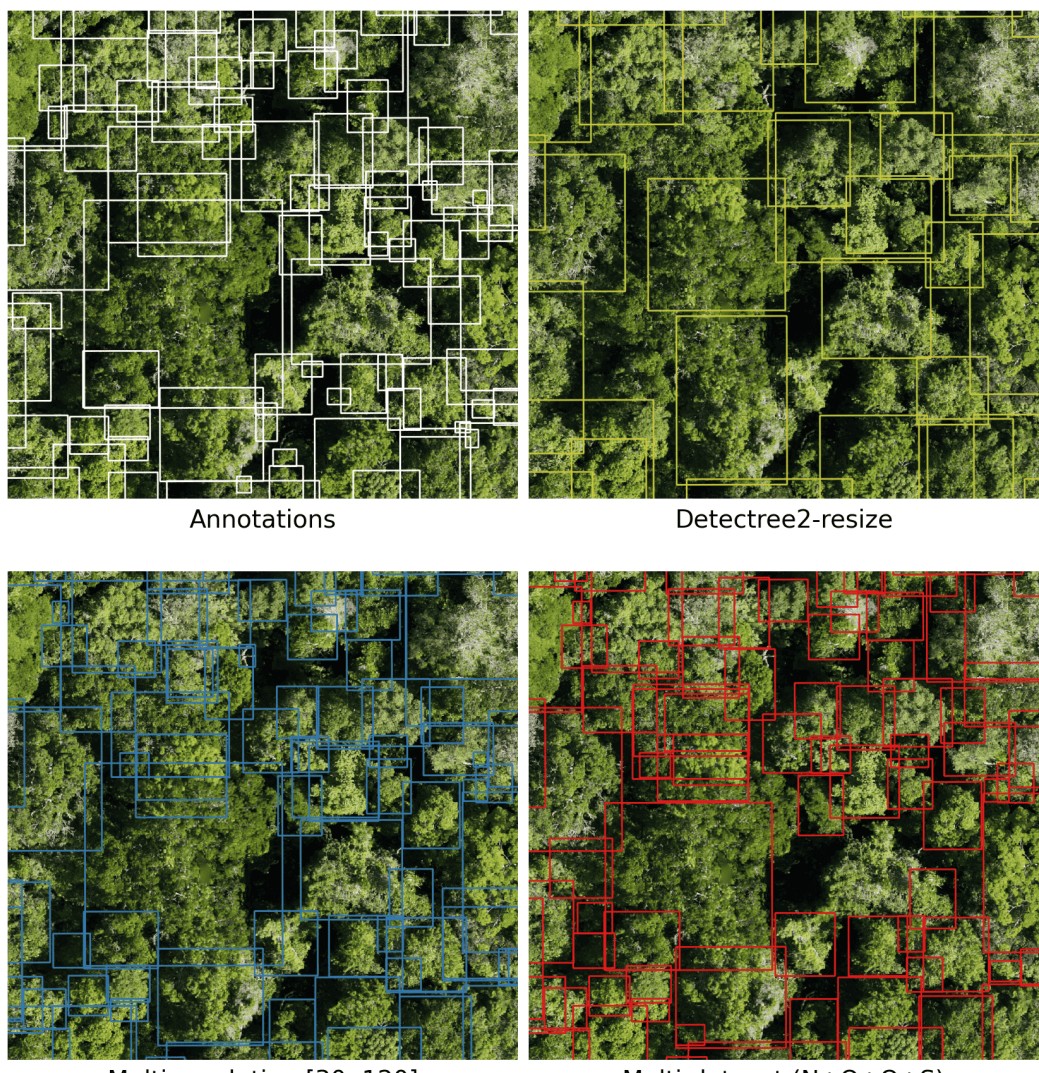

Figure 14: **Qualitative results on SELVABOX (Brazil)**. We compare the annotations in white, the best competing method Detectree2-resize (OOD) in yellow, our best multi-resolution [30, 120] model (ID) in blue and our best model trained on multi-dataset + SELVABOX (ID) in red. Results are shown post-NMS, using the optimal NMS IoU ($\tau_{\mathrm{nms}}$) and score ($s_{\mathrm{min}}$) thresholds for RF1$_{75}$ from Algorithm 1 (see Section B.3 for exact values).

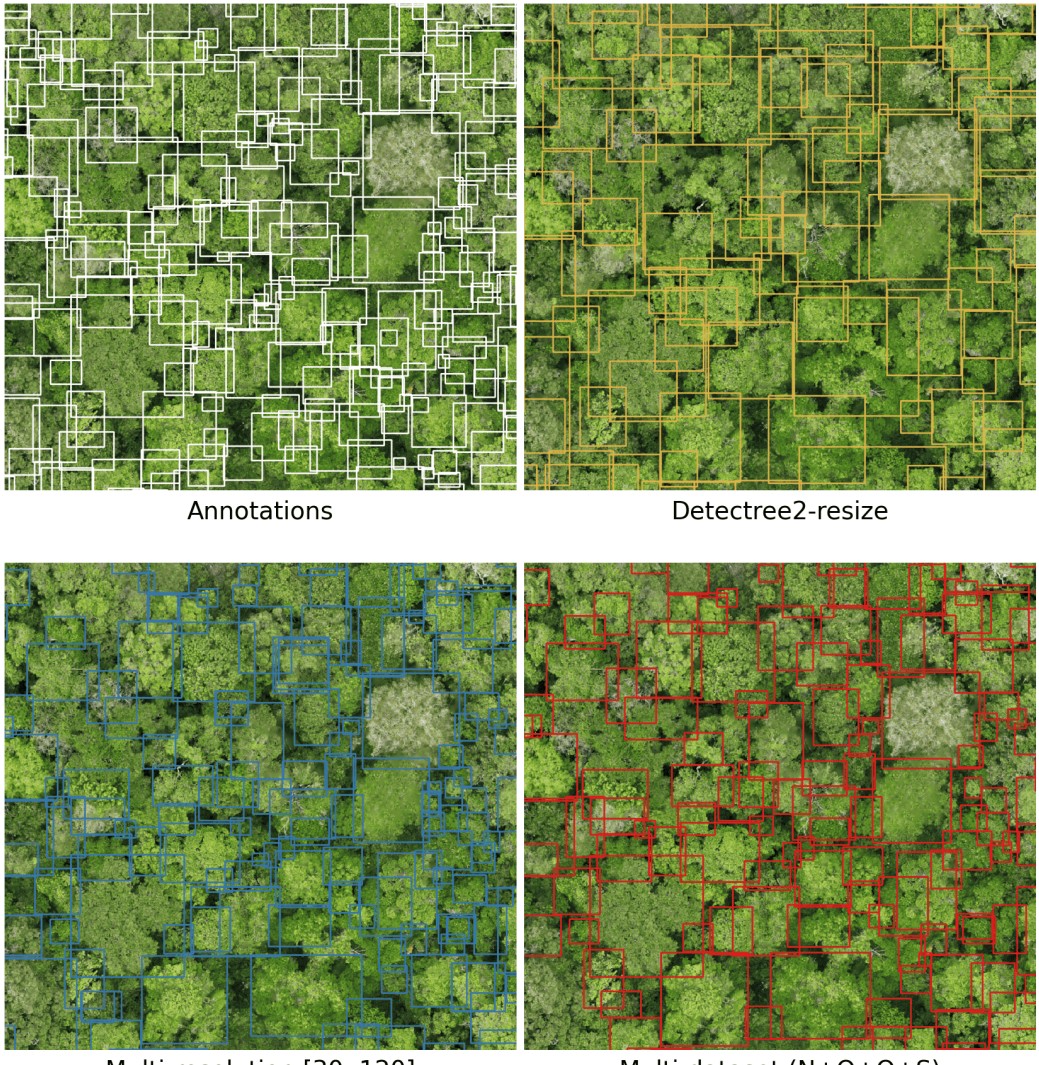

Figure 15: **Qualitative results on SELVABOX (Ecuador)**. We compare the annotations in white, the best competing method Detectree2-resize (OOD) in yellow, our best multi-resolution [30, 120] model (ID) in blue and our best model trained on multi-dataset + SELVABOX (ID) in red. Results are shown post-NMS, using the optimal NMS IoU ($\tau_{\mathrm{nms}}$) and score ($s_{\mathrm{min}}$) thresholds for RF1$_{75}$ from Algorithm 1 (see Section B.3 for exact values).

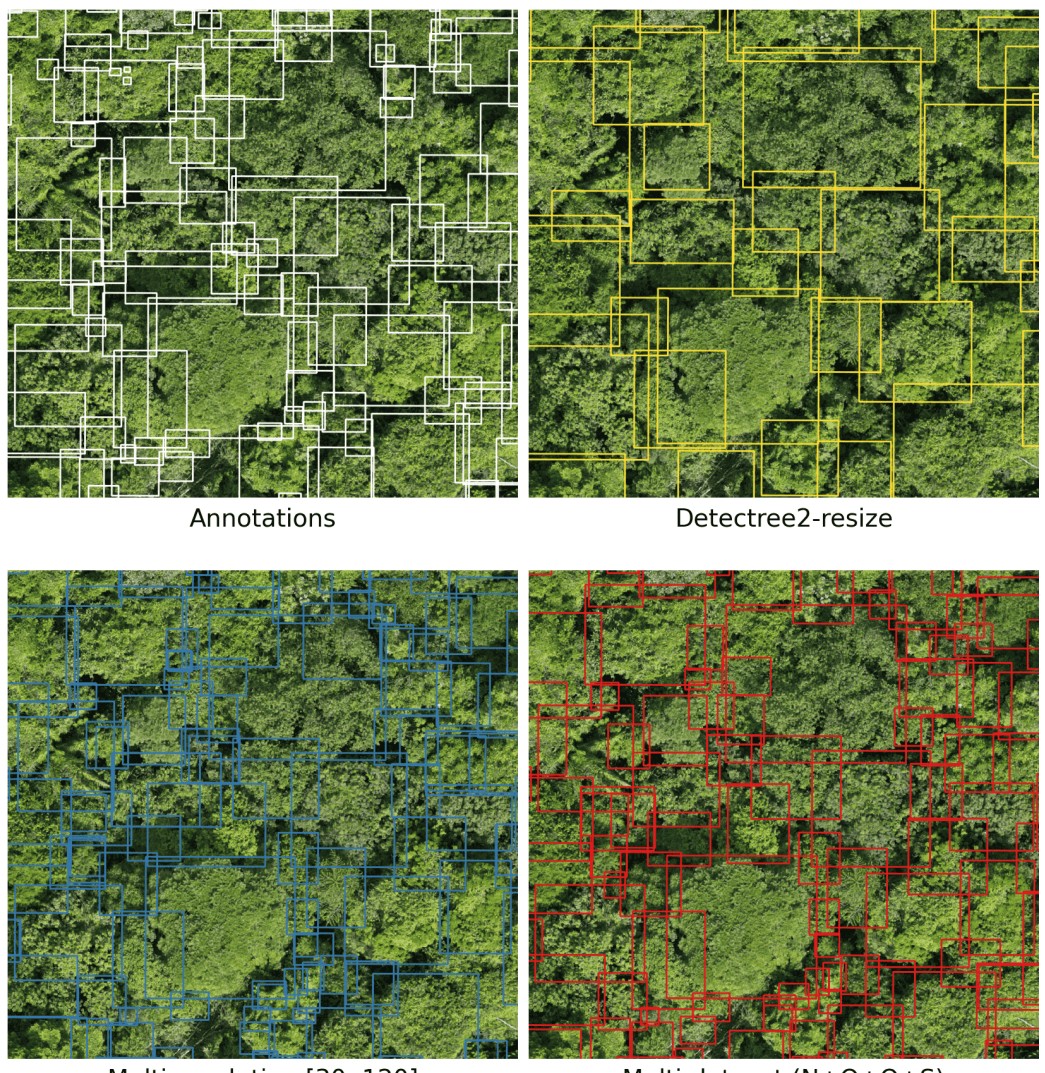

Figure 16: **Qualitative results on SELVABOX (Panama)**. We compare the annotations in white, the best competing method Detectree2-resize (OOD) in yellow, our best multi-resolution [30, 120] model (ID) in blue and our best model trained on multi-dataset + SELVABOX (ID) in red. Results are shown post-NMS, using the optimal NMS IoU ($\tau_{\mathrm{nms}}$) and score ($s_{\mathrm{min}}$) thresholds for RF1$_{75}$ from Algorithm 1 (see Section B.3 for exact values).

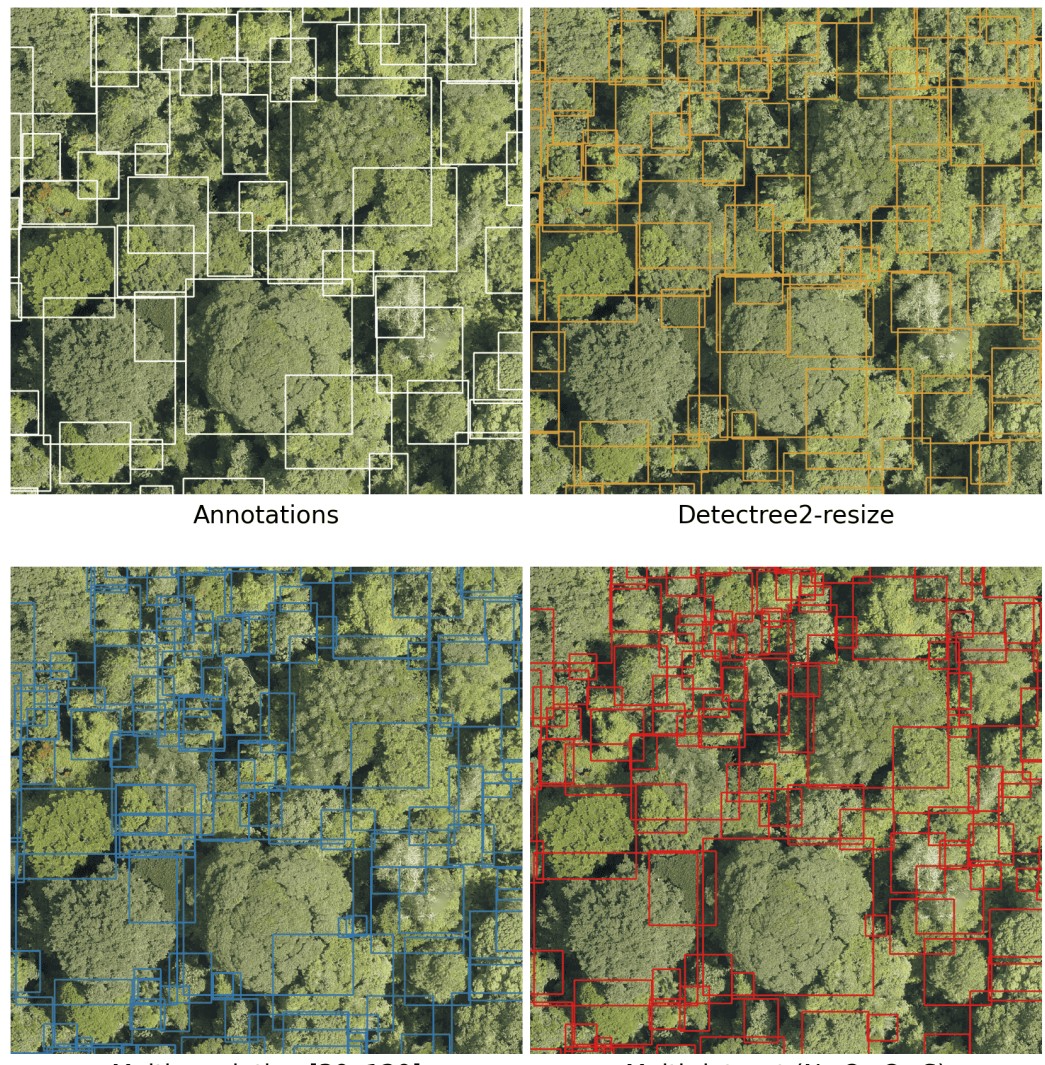

Figure 17: **Qualitative results on BCI50ha**. We compare the annotations in white, the best competing method Detectree2-resize (OOD) in yellow, our best multi-resolution [30, 120] model (OOD) in blue and our best model trained on multi-dataset + SELVABOX (OOD) in red. Results are shown post-NMS, using the optimal NMS IoU ($\tau_{\mathrm{nms}}$) and score ($s_{\mathrm{min}}$) thresholds for RF1$_{75}$ from Algorithm 1 (see Section B.3 for exact values).

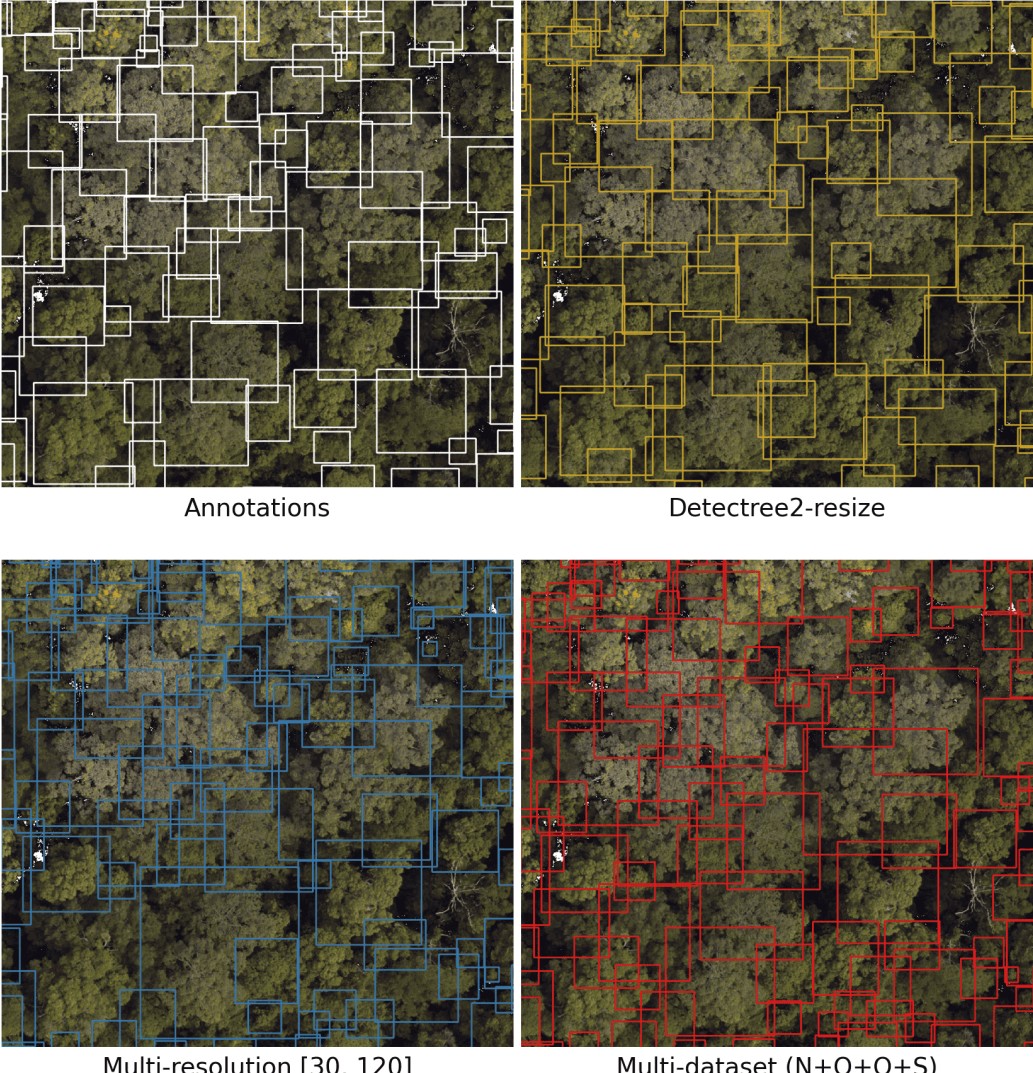

Figure 18: **Qualitative results on Detectree2 dataset**. We compare the annotations in white, the best competing method Detectree2-resize (ID; possibly affected by train–test leakage, since we couldn't recover their data splits) in yellow, our best multi-resolution [30, 120] model (OOD) in blue and our best model trained on multi-dataset + SELVABOX (OOD) in red. Results are shown post-NMS, using the optimal NMS IoU ($\tau_{\mathrm{nms}}$) and score ($s_{\mathrm{min}}$) thresholds for RF1$_{75}$ from Algorithm 1 (see Section B.3 for exact values).

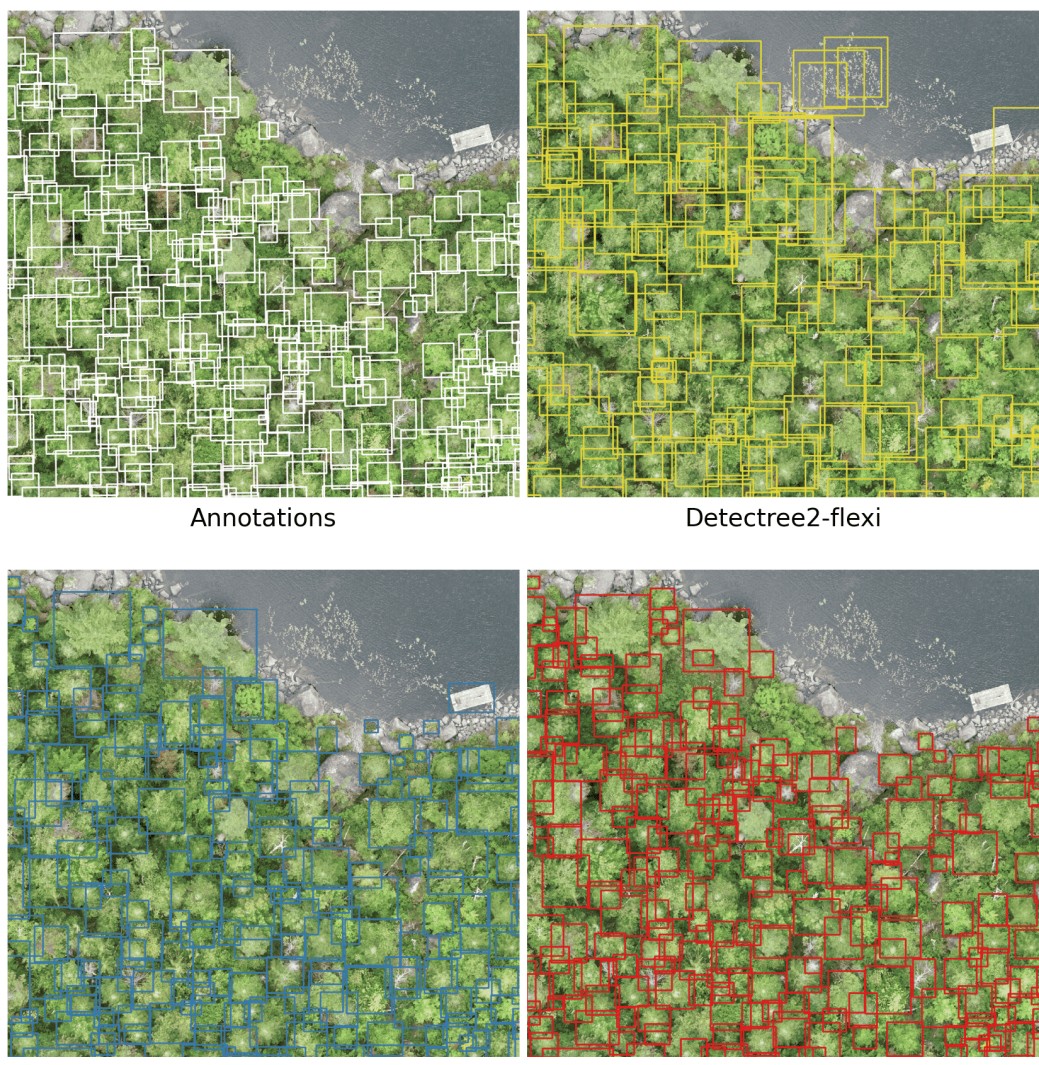

Figure 19: **Qualitative results on QuebecTrees**. We compare the annotations in white, the best competing method Detectree2-flexi (OOD) in yellow, our best multi-resolution [30, 120] model (OOD) in blue and our best model trained on multi-dataset + SELVABOX (ID) in red. Results are shown post-NMS, using the optimal NMS IoU ($\tau_{\text{nms}}$) and score ($s_{\text{min}}$) thresholds for RF1$_{75}$ from Algorithm 1 (see Section B.3 for exact values).

## G    PYTHON LIBRARIES

### G.1    GEODATASET

We've released our pip-installable Python library *geodataset* on GitHub under the permissive Apache 2.0 license. The library serves four main purposes: ① Tilerizers for cutting rasters into tiles—with resampling, AOI, and pixel-masking support—for training/evaluation (as COCO-style JSON) or inference; ② an Aggregator tool that converts predicted object coordinates back into the original CRS and efficiently performs NMS on large sets of detections (at the raster-level); ③ base dataset classes for training and inference that integrate easily with PyTorch's DataLoader; and ④ standardized conventions for naming tiles and COCO JSON files. See the repository documentation for more details.

### G.2    CANOPYRS

We've released a Python GitHub repository called *CanopyRS* to replicate our results, benchmark models, and infer on new forest imagery. It's distributed under the permissive Apache 2.0 license and leverages *geodataset* for pre- and post-processing, with Detectron2 and Detrex handling model training. Its modular design makes it easy to extend in future work—for example, supporting instance segmentation, clustering, or classification of individual trees. See the repository documentation for more details.

## H    USE OF LARGE LANGUAGE MODELS (LLMS)

We used LLMs as general-purpose assistive tools to improve the clarity and conciseness of the text, as well as for occasional coding assistance. These tools were not used for research ideation or to generate novel scientific content. All conceptual and experimental contributions were made by the authors.

