# OpenReview forum: "SelvaBox: A high‑resolution dataset for tropical tree crown detection"
_ICLR.cc/2026/Conference — ICLR 2026 Poster_

### Official Review · Reviewer_mwqm · 2025-10-23

**Soundness:** 3
**Presentation:** 4
**Contribution:** 4
**Rating:** 8
**Confidence:** 4

**Summary:**

The paper introduces SELVABOX, a high-resolution dataset for tropical tree crown detection, featuring over 83,000 manually labeled crowns from drone imagery across Brazil, Ecuador, and Panama. It highlights the challenges of detecting individual tree crowns in tropical forests due to their variability and overlap, advocating for advanced remote sensing methods.

**Strengths:**

# Strengths:

(+) The SELVABOX dataset's scale, with over 83,000 annotations, is an order of magnitude larger than existing tropical forest datasets, providing a substantial resource for training and validation.

(+) Higher-resolution inputs consistently enhance detection accuracy, as demonstrated through extensive benchmarks, offering valuable insights for model optimization.

(+) The open-access nature of the dataset, code, and pre-trained weights fosters collaboration and accelerates research in tropical forest monitoring.

**Weaknesses:**

# Weakness:

(-) Incomplete annotations in some areas (e.g., Brazil and Ecuador rasters) may introduce noise and reduce model reliability, despite the masking strategy employed.

(-) The reliance on RGB-only validation, due to logistical and cost constraints, may limit accuracy compared to LiDAR-based methods in complex tropical canopies.

(-) The paper lacks a detailed comparison with LiDAR-based datasets, potentially overlooking a key alternative for tropical forest monitoring.

Minor Suggestions:

Some prior works of detection in forest scenarios are missing [1-5]. It would be better if the author could discuss them in the related work.

**Ref:**

[1] From drones to phenotype: using UAV-LiDAR to detect species and provenance variation in tree productivity and structure. Remote Sensing, 2020.

[2] Explainable identification and mapping of trees using UAV RGB image and deep learning. Scientific Reports, 2021.

[3] A general deep learning model for bird detection in high-resolution airborne imagery. Ecological Applications, 2022.

[4] Benchmarking wild bird detection in complex forest scenes. Ecological Informatics, 2024.

[5] Enhancing palm precision agriculture: An approach based on deep learning and UAVs for efficient palm tree detection. Ecological Informatics, 2025.

**Questions:**

# Questions:

1. How does the masking strategy for incomplete annotations impact model training performance compared to datasets with full annotations?

2. What are the specific challenges of scaling RGB-only methods to diverse tropical forest conditions not addressed by SELVABOX?

3. How might the inclusion of LiDAR data enhance the detection accuracy of models trained on SELVABOX?

4. What are the potential biases introduced by the manual annotation process conducted by a limited number of biologists?

5. How does the zero-shot performance of SELVABOX-trained models compare to fine-tuned models on non-tropical datasets over a longer term?

---

> ### Author Response · Authors · 2025-11-21
>
> We thank the reviewer for the constructive feedback and the positive comments on the scale of the SelvaBox dataset, the empirical findings on the benefits of higher-resolution inputs for detection accuracy, and the open-access release of the dataset, code, and pre-trained weights for tropical forest monitoring. Below, we address each concern point by point and indicate the corresponding changes in the revised manuscript (changes are highlighted in blue).
>
> ### W.1/Q.1. Masking strategy:
>
> We acknowledge that incomplete annotations may influence model training; however, our masking strategy is specifically designed to avoid treating unlabeled trees as negatives (see our answer to W.2/Q.2. of reviewer qCUJ). This ensures training focuses on regions where annotations are present and reliable. To different degrees, every tree detection dataset is subject to the same kind of issues. For example, while OAM-TCD has an impressive geographic coverage [1], they employed non-biologist annotators (raising questions about the completeness of annotations) and the dataset contains tree cluster annotations that we had to mask during processing to avoid clashes with single-tree annotations.
>
> However, we note that QuebecTrees gives a reasonable point of reference: biologists surveyed plots on foot and tried to systematically annotate the crown of every tree whose crown was visible in the imagery, which is easier in temperate forests. Based on that, we can compare results from Table 5: our methods trained only on SelvaBox (S) achieve significantly better results on QuebecTrees in an OOD setting compared to competing methods trained on other datasets (Detectree2 variants and DeepForest (N)). This is a strong indication that the masking strategy we employed on SelvaBox is not a significant issue here.
>
> Additionally, comparing our DINO-Swin-L (NQO) and (NQOS) models, we observe that they are within standard deviation of each other on every non-tropical dataset evaluated (NeonTreeEvaluation, QuebecTrees, and OAM-TCD). Since the only difference between these models is the inclusion of SelvaBox, we conclude that the masking strategy prevents SelvaBox’s annotation noise from negatively impacting performance on other datasets.
>
> While extensive ablation studies varying annotation masking proportions (for example, on the QuebecTrees dataset) could be a valuable direction for future work to strictly quantify the impact of annotation completeness, the empirical evidence presented above suggests that our current strategy is robust.
>
> [1] Veitch-Michaelis et al., OAM-TCD: A globally diverse dataset of high-resolution tree cover maps. In NeurIPS 2024.
>
> ### Q.2. Scaling RGB-only methods:
>
> We identify three main categories of challenges in scaling RGB-only methods that are not fully addressed by SelvaBox.
>
> **1. Operational reality**: Scaling RGB drone campaigns across the tropics requires sustained local capacity and infrastructure. In practice, work in these regions faces significant meteorological constraints (frequent rain and low-altitude fog) and sometimes complex permitting hurdles. Furthermore, capacity building is resource-intensive; training local collaborators is essential to build respectful and long-lasting relationships and ensure local priorities are met, but this process takes significant time and financing.
>
> **2. Domain Shifts across Tropical Forests**: SelvaBox is currently Neotropical-centric. Scaling to Paleotropical forests (e.g., Southeast Asia, Central Africa) introduces domain shifts in species composition and crown architecture that our current data do not fully capture. Therefore, models trained on SelvaBox cannot be assumed to generalize seamlessly to all tropical regions and phenological states. However, we note that our DINO-Swin-L (NQOS) model performed close to the in-distribution competing method (Detectree2) on their dataset in Malaysia (Table 4), indicating that the model retains significant robustness despite these shifts.
>
> **3. Limited access to vertical structure from RGB**: A main challenge with detecting trees from RGB drone imagery is that RGB is restricted to emergent and upper canopy trees because the RGB signal does not penetrate through dense canopies, unlike LiDAR which partially penetrates foliage. However, the emergent and upper canopy trees that can be detected from RGB drone imagery are foundational to forest ecosystems since they store about 70% of the tree biomass. While LiDAR could help to bring additional information about understorey forest structure, LiDAR is considerably more expensive than RGB imagery, and thus out of reach of many tropical scientists.

---

> ### Author Response · Authors · 2025-11-21
>
> ### W.2-3/Q.3. LiDAR applications:
>
> We agree with the Reviewer that LiDAR data could enhance detection accuracy of models trained on SelvaBox and other RGB datasets in some cases. LiDAR provides additional structural information that could improve performance in two ways: (1) annotators can leverage point clouds to help distinguish trees in some challenging cases (e.g., overlapping crowns from the same species, crowns with similar heights that cannot be distinguished with RGB+DSM), potentially yielding higher-quality annotations, and (2) models trained on multimodal data could learn to exploit structural details absent in RGB alone, such as vertical structure or stem locations.
>
> While LiDAR excels at detecting trees beneath the canopy, regions invisible to RGB, we emphasize that RGB photogrammetric point clouds generated from drone structure-from-motion and orthomosaics offer a cost-effective alternative for detecting emergent and upper canopy trees, making this research more accessible to local communities and ecologists across the globe. Our ongoing work on multimodal RGB and point cloud fusion directly addresses this direction and is beyond the scope of the present study.
>
> ### Q.4. Potential biases in manual annotations:
>
> Despite substantial effort to ensure annotation quality—including expert annotator training, multi-pass review, and systematic quality control (Appendix A.3)—our dataset may reflect inherent human biases.
>
> First, annotators may disagree on ambiguous cases: while larger tree crowns are consistently annotated, it is sometimes unclear whether branches adjacent to a crown belong to the same tree or represent independent understory trees.
> Similarly, distinguishing smaller, heavily intertwined crowns is challenging, and annotators may apply varying difficulty thresholds when deciding whether an object constitutes an actual tree versus a liana or other vegetation.
>
> Additionally, biologists may differ in precision when drawing bounding boxes around tree crowns, with some systematically including more padding than others. To mitigate these issues, annotators were asked to discuss difficult cases and reach consensus. None of these concerns were flagged as major during our systematic quality control. We will integrate these additional details in Appendix A.3.
>
> As discussed in our answer to Q.3, engaging LiDAR experts and additional biologist annotators could further reduce these biases; however, substantial additional funding would be necessary to maintain dataset scale. We believe our annotation protocol effectively limited these biases, which have minimal impact on SelvaBox’s overall quality and on model performance, as illustrated by the qualitative results in the Appendix.

---

> ### Author Response · Authors · 2025-11-21
>
> ### Q.5. Zero-shot and fine-tuned models comparison:
> We understand this question as asking how a detector trained only on SelvaBox performs in zero-shot mode compared to models fine-tuned on each target dataset. In our experiments, we evaluate models at a fixed point in time and do not study temporal evolution.
>
> To answer this question, let’s consider evaluation results in Table 5 on NeonTreeEvaluation, QuebecTrees and OAM-TCD (all non-tropical datasets). We first observe that our DINO-Swin-L (S) method trained on SelvaBox alone performs poorly on NeonTreeEvaluation (N) at 5.16 mAP compared to 23.68 mAP for our best in-distribution method trained on this dataset alone (DINO-Swin-L (N)). This is expected given the domain shift and shows that fine-tuning on the target dataset clearly outperforms a purely zero-shot SelvaBox model. However, our multi-dataset DINO-Swin-L variants (N+Q+O) and (N+Q+O+S) are within standard deviation of each other on NeonTreeEvaluation, suggesting that adding SelvaBox to a non-tropical mix of training datasets does not weaken performance on NeonTreeEvaluation despite differences in geographic location, GSD and sensor. Although they do not significantly improve the results on this dataset, our variant methods show great generalization capacities considering the evaluation over all datasets, either in-distribution or out-of-distribution.
>
> On QuebecTrees (Q) and OAM-TCD (O) datasets, our DINO-Swin-L (S) method trained on SelvaBox alone outperforms the competing methods Detectree2 and DeepForest by a significant margin. Similarly to our discussion on NeonTreeEvaluation (N) in the previous paragraph, DINO-Swin-L (N+Q+O) and DINO-Swin-L (N+Q+O+S) models achieve very similar scores, with differences within standard deviation. This indicates that adding SelvaBox (S) to the non-tropical training mix does not degrade performance on QuebecTrees or OAM-TCD, despite the distribution shifts (tropical vs non-tropical and/or urban).
>
> Together, these in-distribution and out-of-distribution evaluations highlight the quality of SelvaBox as a standalone training dataset or within a multi-dataset framework, including when models are evaluated on non-tropical datasets. We hope these clarifications address Q.5, and please let us know if you have further questions.
>
> ### Minor suggestions:
>
> We thank the Reviewer for suggesting these additional references, which we will integrate in the corresponding paragraphs of the revised manuscript when applicable.

---

> ### Comment · Reviewer_mwqm · 2025-11-21
> **Reply**
>
> Thank you for your detailed response. All my concerns are well addressed. I’ll keep my rating, good luck.

---

### Official Review · Reviewer_qCUJ · 2025-10-26

**Soundness:** 3
**Presentation:** 3
**Contribution:** 3
**Rating:** 6
**Confidence:** 4

**Summary:**

This paper introduces **SELVABOX**, a large-scale UAV RGB dataset for detecting individual tree crowns in tropical forests across Brazil, Ecuador, and Panama. The authors also propose a practical evaluation metric, **RF175**, which measures raster-level F1 at an IoU threshold of 0.75. The dataset contains more than 83k high-quality crown annotations and comes with a complete processing pipeline, open-source code, and pretrained models.
Extensive experiments are conducted with both CNN and Transformer detectors. The results show that higher image resolution and multi-resolution training can significantly improve generalization, even across different domains. Overall, the dataset and benchmark make a timely and meaningful contribution to ecological AI and forest monitoring.

**Strengths:**

- **High-impact dataset:**  SELVABOX fills an important gap for tropical regions. The scale, diversity, and annotation detail are impressive, and the dataset is likely to become a strong reference for future research in this area.
- **Practical evaluation metric:**  The proposed RF175 metric focuses on raster-level performance rather than patch-level mAP, which makes sense for real-world forestry applications.
- **Comprehensive experiments:**  The authors benchmark various models, image resolutions, and training setups, providing clear evidence for each design choice.
- **Good open-science practice:**  The dataset, preprocessing pipeline, and code are fully released, supporting reproducibility and future extensions.
- **Clear and well-written paper:**  The paper is well organized, the figures are helpful, and the results are easy to interpret.

**Weaknesses:**

1. The paper does not report any **quantitative analysis of annotation consistency**. Although the annotation process is described in detail, we do not know how consistent different annotators were. Some inter-annotator agreement numbers (like IoU or Cohen’s kappa) would really help readers trust the data quality.
2. The **test set completeness** is mentioned but not clearly verified. It would be good to show that the test areas were fully annotated or to quantify potential missing crowns. Without this, the reliability of RF175 as a “gold-standard” metric is slightly uncertain.
3. The **RF175 metric** seems quite sensitive to the IoU threshold and NMS parameters, yet this sensitivity is not explored. It would be interesting to see how performance changes for IoU thresholds of 0.6 or 0.8.
4. The **Detectree2 evaluation** may include unintentional data overlap because its original split cannot be reproduced. The authors mention this but still treat it as an “in-distribution” case, which could confuse readers. A clearer statement or a controlled subset would make the comparison more trustworthy.
5. The **comparison with existing datasets** is mostly visual. Statistical comparisons, such as KL divergence or KS tests on crown-size distributions, would make the argument for SELVABOX’s uniqueness stronger.
6. The **ethical and dual-use discussion** is too brief. The dataset could be misused, for example, for illegal deforestation or land exploitation. A short section on “Responsible Use” with clear terms or usage guidelines would be helpful.

**Questions:**

1. Could you share some statistics on **annotation consistency**? Even small samples of double-annotated areas or agreement rates would help quantify data reliability.
2. For the **test set completeness**, did you perform any internal audit or random visual checks? If so, please describe how it was done.
3. Have you tried testing **RF175 under different IoU thresholds** (e.g., 0.6, 0.7, 0.8)? A sensitivity curve would be nice to include, even in the appendix.
4. Regarding the **Detectree2 benchmark**, is there a chance of overlapping geographic regions? It might help to clarify this in the final version and mention it clearly as a limitation.
5. Could you add a small **quantitative comparison with other datasets**, perhaps comparing crown-size or bounding-box distributions numerically instead of only visually?
6. The paper would benefit from a short **Responsible Use or Ethics** paragraph in the main text. You could briefly outline intended use, potential misuse, and the licensing policy.
7. It might also help to add **one or two failure case visualizations**, showing where the models still struggle (e.g., dense canopy overlap or mixed-species confusion). This would make the benchmark even more insightful.
8. Have you considered adding **statistical confidence intervals** (like standard deviations over random seeds) for RF175 in the main tables? It would make the reported numbers more interpretable.
9. Some references could be considered in Introduction and Related works: such as 10.1109/MGRS.2024.3479871, https://doi.org/10.1016/j.rse.2023.113485 and https://doi.org/10.1016/j.isprsjprs.2020.07.002

---

> ### Author Response · Authors · 2025-11-21
>
> We thank the reviewer for the constructive feedback and the positive comments on the impact and scale of SelvaBox for tropical forests, the practicality of our RF1_75 metric, the comprehensiveness of our experimental study, the open release of the dataset and code, and the clarity and organization of the paper. Below, we address each concern point by point and indicate the corresponding changes in the revised manuscript (changes are highlighted in blue).
>
> ### W.1/Q.1. Annotation consistency:
>
> As we describe in Section 3 and Appendix A.3, our annotation procedure is such that for each orthomosaic, a first initial round of manual annotations is produced. Then, each annotated region is reviewed at least once following the procedure described in App. A.3. We would like to clarify that the reviewing step, in which missing annotations are added and existing ones potentially refined, is a validation of the annotations. We did not ask different annotators to annotate the same region from scratch. Nonetheless, emphasis was put on producing  high-quality annotations and the annotation process represented 1,284 people-hours of work. The annotation revisions were produced on the same ArcGIS Pro layer as the initial annotations, and the software does not have a feature for keeping the change history. Therefore, we are not able to provide an estimation of the number of boxes that were modified by the reviewers, or the number of added boxes in the reviewing step(s).
> While we cannot provide a quantification of inter-annotator agreement, we believe that annotations are consistent across all sites because all annotations have been validated by two annotators. The annotators also consulted each other when they were unsure about certain annotations. Moreover, as highlighted by Reviewer QBSE, we believe that the consistently greater or equivalent cross-dataset performance of models trained with SelvaBox demonstrates the quality of the annotations.
>
> ### W.2/Q.2. Test set completeness:
>
> We designed a systematic splitting pipeline with manual visual checks to define train, valid and test splits while ensuring the completeness of the test set. One may note this pipeline is independent and comes after the annotations protocol presented in Appendix A.3. The splitting pipeline procedure is as follows.
>
> **Step 1**: For each raster we defined an initial area-of-interest (AOI) polygon that removes raster borders where imagery quality drops due to artifacts and where annotations stop. We ensured a small padding between annotations and the AOI border so as not to cut through annotated crowns, while avoiding large margins that would leave entire unannotated trees inside the AOI and thus introduce false negatives.
>
> **Step 2**: We visually scanned the rasters and created AOI “holes” in regions where annotations were too scarce. These holes were then used to automatically mask the corresponding pixels and, in a few cases, standalone annotations overlapping these holes. This step effectively pruned regions where missing annotations would be most problematic.
>
> **Step 3**: For each of the three countries in SelvaBox (Brazil, Ecuador and Panama), we selected the highest-quality AOI regions as test-set candidates, i.e., zones with no AOI holes and no significant imagery reconstruction artifacts.
>
> **Finally**, we balanced criteria when finalizing the test split: we aimed for approximately 10–15% of the AOI area in the test set for each country, while ensuring diversity in forest types and contexts (primary, secondary and plantation forests, presence of water bodies such as lakes and rivers, etc.). In an ideal scenario, entire rasters would be dedicated to the test split; however, we found that the largest rasters were often those with the highest imagery and annotation quality, and assigning them entirely to the test split would not leave enough data for training. We therefore reserved subregions of these rasters for the test split, ensuring that test AOIs were as complete as possible, which was our main criterion.
> We then repeated steps 2 and 3 for the validation split and assigned the remaining AOI regions to the training split (only 50% of the training split rasters have AOI holes, the rest are of high quality such as test and valid). The resulting AOIs are shown in Appendix A.2.

---

> ### Author Response · Authors · 2025-11-21
>
> ### W.3/Q3. RF1 ablation study:
>
> We thank the reviewer for raising this point, which was not covered in the first version of the paper. We performed an ablation study on RF1 versus IoU threshold for the four datasets on which RF1 can be defined: SelvaBox, BCI50ha, Detectree2 and QuebecTrees (we have raster-level annotations for these datasets). We used IoU thresholds from 0.50 to 0.95 with a step of 0.05 and produced a plot for each dataset, optimizing the NMS parameters independently for each IoU threshold (see Figure 4 and Appendix E in the revised manuscript).
>
> The RF1 curves for SelvaBox and QuebecTrees are consistent with what we expected from Tables 4 and 5: our DINO-Swin variants significantly outperform competing methods (both Detectree2 checkpoints as well as DeepForest (N)) at every IoU threshold. We find that the RF1 curves for BCI50ha and Detectree2 are also interesting: the Detectree2-resize model performs better at an IoU of 0.70 than at 0.65 on Detectree2, and better than at IoU thresholds in [0.50, 0.65] on BCI50ha. This is not what we initially expected, because the performance of a model should generally decrease as the evaluation IoU threshold becomes more restrictive. However, there are two simple explanations for this: (1) we optimized the NMS parameters independently for each IoU threshold, so the model may perform unusually well at particular thresholds, and (2) the BCI50ha and Detectree2 datasets are much smaller than SelvaBox and QuebecTrees (4.7k and 3.8k annotations compared to 83k and 23k, respectively), which may lead to higher variance.
>
> Overall, these insights further underline the quality and scale of SelvaBox, compared to smaller datasets where annotation noise and crown size distribution have a larger impact on metric variance. In future work, it would be interesting to develop an RF1_50:95 metric, analogous to the standard mAP and mAR_50:95, where NMS hyperparameters are optimized against the average RF1 over multiple IoU thresholds on the validation set. We’ve added these new results and limitations to the manuscript in Section 5.3.
>
> ### W.4/Q.4. Detectree2 evaluation:
>
> As noted in L.424, the Detectree2 authors did not release their original train-test splits or provide documentation regarding data partitioning. Consequently, we cannot ensure rigorous separation when evaluating their model on this dataset, which limits the interpretability of comparative results.
>
> To address this limitation transparently, we revise L.424 to state: '...during evaluation on their dataset, limiting the interpretation of the results...' We also update Table 4 by replacing the "X*" marker with "~" in the OOD column and adding the clarification: 'We denote with ~ the Detectree2 competing methods where original train-test splits could not be recovered, preventing controlled evaluation on their dataset and limiting the interpretability of comparative results.'
>
> Notably, despite treating the Detectree2 dataset as fully out-of-distribution in all other experiments, our method achieves comparable performance to Detectree2 models that benefited from this data leakage, underscoring the robustness of our approach.

---

> ### Author Response · Authors · 2025-11-21
>
> ### Q.5. Statistical comparison with other datasets:
>
> To rigorously compare SelvaBox with existing datasets as suggested by the Reviewer, we quantified crown-size distribution differences using three complementary statistical approaches:
>
> **Methodology:** We computed the Jensen-Shannon (JS) distance and Kullback-Leibler (KL) divergence across datasets. Since KL divergence is asymmetric and unbounded, making interpretation difficult, we prioritize JS distance, which is symmetric, bounded between 0 and 1, and well-suited for discrete distributions. We also performed two-sample Kolmogorov-Smirnov (KS) tests comparing SelvaBox against existing datasets. The KS test evaluates the maximum vertical distance between empirical cumulative distribution functions (ECDFs), providing a non-parametric, distribution-free measure robust to differences in dataset size. Given that KS test p-values follow $p \approx 2e^{-2 n \cdot D^2}$ [1], where $n$ is dataset size and $D$ the KS statistic, and our dataset sizes range from $n = 3,947$ (Detectree2) to $266,663$ (OAM-TCD), we expect p-values approaching $1.04\cdot 10^{-35}$ when $D \ge 0.1$ and  $n \ge 3,947$. We improved Figure 12 (Appendix F.2) to detail crown-size distributions for each dataset.
>
> |||BCI50ha|Detectree2|NeonTreeEvaluation|QuebecTrees|OAM-TCD|
> |-|-|-|-|-|-|-|
> |SelvaBox|JS Distance|0.6248|0.6476|0.2789|0.2622|0.2152 |
> |SelvaBox|KL Divergence|2.4293|2.1779|0.3826|0.7028|0.1687|
> |SelvaBox|KS Test|0.6231|0.6338|0.3257|0.2270|0.2117|
> |SelvaBox|KS Test p-value|$<1 \cdot 10^{-35}$|$<1 \cdot 10^{-35}$|$<1 \cdot 10^{-35}$|$<1 \cdot 10^{-35}$|$<1 \cdot 10^{-35}$|
>
> **Results:** SelvaBox exhibits substantially different crown-size distributions from tropical datasets (JS Distance: BCI50ha = 0.6248, Detectree2 = 0.6476) and moderately different distributions from temperate datasets (NeonTreeEvaluation = 0.2789, QuebecTrees = 0.2622). Pairwise KS tests reveal highly significant differences across all comparisons (p < 0.001, KS statistics ranging from 0.2117 to 0.6338), confirming that SelvaBox's crown-size distribution is statistically distinct from all existing datasets. The large annotation counts (3,947 to 266,663 samples) ensure these distributional differences are meaningful and robust to dataset scale variations.
>
> **Interpretation:** OAM-TCD shows the greatest similarity to SelvaBox (JS Distance = 0.2152), likely because its large geographic scale and multi-biome coverage encompass diverse crown morphologies, unlike datasets restricted to single regions or biomes. The asymmetric KL divergence values further support these conclusions, demonstrating how SelvaBox uniquely captures the crown-size distributions and structural diversity of tropical forests.
> We add these discussions and new results in the revised manuscript in Appendix F.2.
>
> [1] Marsaglia et al, Evaluating Kolmogorov's Distribution. In Journal of Statistical Software 2003.
>
>
> ### Q.6. New paragraph:
> We agree that the paper would benefit from a short Responsible Use / Ethics paragraph. In the revised version, we add a section 6 that briefly outlines intended use, potential misuse, and our licensing policy for both the dataset and the released models.

---

> ### Author Response · Authors · 2025-11-21
>
> ### Q.7. Failure qualitative results:
>
> While we did not add new visual examples of failure cases, we direct the Reviewer's attention to the qualitative results provided in Appendix F.5 and F.6, which illustrate method limitations across diverse datasets through representative failure cases. Notable examples include the following:
>
> **Overlapping and intertwined crowns**: In Figure 15 (SelvaBox-Ecuador), both our Multi-resolution and Multi-dataset methods fail to detect the trees in the bottom-left corner of the image. This appears counter-intuitive, as both models are in-distribution, whereas the out-of-distribution Detectree2 performs better in this specific region. However, closer inspection reveals that the canopy is highly ambiguous, with multiple overlapping crowns of similar color and texture. Our DINO methods lack sufficient confidence in these ambiguous detections, and the predictions are filtered out by the validation-derived score thresholds (0.40 and 0.50) and NMS. In contrast, Detectree2 predictions are retained because its optimal score threshold, determined during validation, is substantially lower (0.20), likely because the model performs worse overall on SelvaBox (see Table 4) and requires a lower confidence bar to register detections.
> In Figure 17 (BCI50ha), all evaluated models fail to accurately detect the 2 trees in the top-right corner of the tile. This represents the challenge of intertwined canopy crowns, where accurate detection of individual crown instances is difficult.
>
> **Detecting other objects**: In Figure 19 (QuebecTrees), Detectree2-flexi incorrectly
>  detects aquatic plants and rocks, while our Multi-Resolution [30, 120] model incorrectly
>  detects what seems to be a boat or a wooden box in the water. Notably, our fully in-distribution model (NQOS) does not predict rocks or water-based objects, which is expected and demonstrates the robustness gained from diverse training data.
>
> ### Q.8. Standard deviations:
>
> Our main tables (2, 3, 4 and 5) report standard deviations over three seeds for each model that we trained ourselves (DINO-Swin and Faster R-CNN models, as well as every DeepForest variant except for (N)), including RF1_75. However, as you noted, we do not report standard deviations for Detectree2 and DeepForest (N) because we rely on the checkpoints released in their repositories, which do not provide multiple seeds. We will clarify this point in the text to make the presence and absence of variability estimates more explicit.
>
>
> ### Q.9. References:
>
> We thank the Reviewer for suggesting these additional references, which we will integrate into the Introduction section of the revised manuscript.

---

> > ### Comment · Reviewer_qCUJ · 2025-11-22
> > **Thanks for your efforts!**
> >
> > According to the authors' responses and other reviewers' comments, I raise my score to 8.
> > Good luck and looking forward to publish your paper.

---

### Official Review · Reviewer_QBSE · 2025-10-27

**Soundness:** 3
**Presentation:** 3
**Contribution:** 2
**Rating:** 6
**Confidence:** 3

**Summary:**

This paper introduces a dataset, SelvaBox, containing UAV orthoimagery over several neotropical forest locations, covering 500 ha and containing 83k manually annotated tree crowns in the form of bounding boxes. Although larger datasets exist for temperate zones, this dataset aims at providing a large, diverse and high-quality crown detection benchmark for the tropics, with a focus on natural forests. The results show that using SelvaBox along with a transformer-based object detection model leads to state-of-the-art results on several other benchmarks. The paper also proposes an evaluation protocol, includinf the RF1_75 metric computer at the whole raster level as a better way to assess model performance.

**Strengths:**

- The paper is clearly written and well structured.
- The motivation is clear and well formulated.
- The experiments are comprehensive and highlight the usefulness of the new dataset.
- The choices in terms of dataset creation and models are well justified.

**Weaknesses:**

1. This work is quite complete. However, it all relies on the quality of the manual annotations. Although the good cross-dataset performance already indicates that the annotation quality is appropriate (and thus the within-SelvaBox performance does not rely on learning dataset-specific annotation biases), it would still be very helpful to see a quality assessment in the form of inter-annotator agreement analysis.
2. The proposals of new evaluation metric/setup, which is one of the novelty claims in the paper, would be much better supported using such an inter-annotator evaluation, since they would act as an oracle predictor on which the new metrics could be compared to previous ones. Otherwise, there is little evidence that supports the claims about the proposed evaluation protocol.
3. In a way, this paper presents a dataset where the main edge is that it is larger scale than others in the same (tropical forest) setting. As such, there is little novelty to speak of.  Novelty is typically a requirement according to the ICLR reviewer guidelines. I’m not 100% sure what this entails when it comes to datasets, but I would imagine enabling the benchmarking of so far un-benchmarkable tasks. SelvaScope does not allow to evaluate methods on anything that is fundamentally different, although its larger scale and diversity will likely be helpful to train models that will be useful to practitioners. As such, it maybe worth questioning the adequacy of ICLR as a venue to publish this paper, although I do commend the authors for the quality of their work.

**Questions:**

Would it still be feasible to look at inter-annotator agreement? Are some tiles maybe already annotated by two or more experts? Could experts be contacted to do this? I understand that this could be tricky to pull off in the latter case.

---

> ### Author Response · Authors · 2025-11-21
>
> We thank the reviewer for the constructive feedback and the positive comments on the clarity and structure of the paper, the clear motivation, the comprehensiveness of the experiments, the justification of the SelvaBox dataset and model choices, and the overall completeness and practical relevance of the work. Below, we address each concern point by point and indicate the corresponding changes in the revised manuscript (changes are highlighted in blue).
>
> ### W.1/Q.1. Annotation quality assessment:
>
> As we describe in Section 3 and Appendix A.3, our annotation procedure is such that for each orthomosaic, a first initial round of manual annotations is produced. Then, each annotated region is reviewed at least once following the procedure described in Appendix A.3. We would like to clarify that the reviewing step, in which missing annotations are added and existing ones potentially refined, is a validation of the annotations. While we did not ask different annotators to annotate the same region from scratch, each tile has been seen at least twice by annotators/reviewers. Emphasis was put on producing high-quality annotations and the annotation process represented 1,284 people-hours of work.
>
> The annotation revisions were produced on the same ArcGIS Pro layer as the initial annotations, and the software does not have a feature for keeping the change history. Therefore, we are not able to provide an estimation of the number of boxes that were modified by the reviewers, or the number of added boxes in the reviewing step(s). Additional annotations from our expert biologists are no longer feasible within our current timeline.
>
> While we cannot provide a quantification of inter-annotator agreement, we believe that annotations are consistent across all sites because all annotations have been validated by two annotators. The annotators also reported consulting each other when they were unsure about certain annotations. Moreover, as highlighted by the Reviewer, we believe that the consistently greater or equivalent cross-dataset performance of models trained with SelvaBox demonstrates the quality of the annotations.
>
>
> ### W.2. Evaluation protocol:
>
> We acknowledge that inter-annotator agreement would provide a valuable upper bound, i.e. an oracle, representing human-level performance at this resolution. Unfortunately, additional inter-annotator evaluation cannot be completed as explained in our response to W.1.
> However, our evaluation protocol remains appropriate in practice. Raster-level metrics are essential for individual tree crown detection applications [1], addressing concrete operational needs: detecting all trees in a given area while mitigating per-tile artifacts during the evaluation process (edge effects, duplicate detections; see L.144). Our detection protocol prioritizes comprehensive tree detection, targeting a 100% RF1 score on annotated canopy trees.
> We use the RF1_75 metric, which achieves scores of 39–56 on well-annotated datasets and is considered **reliable**, i.e. correlates well with detection of trees of interest, by our collaborators across tropical American, African, and South-East Asian forests. To validate this choice, we conducted an additional ablation study varying IoU thresholds (0.50–0.95) across the four datasets with raster-level annotations (SelvaBox, BCI50ha, Detectree2, QuebecTrees) (see Figure 4 and Appendix E in the revised manuscript). Our DINO-Swin variants consistently outperform competing methods across all IoU thresholds on SelvaBox and QuebecTrees, demonstrating robust performance. Notably, performance variance on smaller datasets (BCI50ha, Detectree2) is substantially higher, highlighting SelvaBox's scale (83k annotations) for reliable metric evaluation. These results validate both our metric selection and the quality of our benchmark.
> While we cannot directly compare against an oracle, our metric choice is validated by domain expertise, operational requirements, and empirical ablation studies. The SelvaBox benchmark establishes a rigorous standard: advancing performance on our high-quality annotations will meaningfully improve tropical forest tree mapping capabilities.

---

> > ### Comment · Reviewer_QBSE · 2025-11-21
> >
> > I thank the authors for their response. It would have indeed been useful to be able to use both annotations in order to get an estimate of the inter-annotator agreement and the added value of the proposed evaluation. I understand that, due to the described conditions, this is not possible. I would therefore be inclined to keep my initial score.

---

> ### Author Response · Authors · 2025-11-21
>
> ### W.3. Dataset novelty:
>
> We appreciate the reviewer's recognition of our work's quality. We respectfully address the novelty concern by highlighting SelvaBox's distinctive contributions:
>
> **Geographic and ecological uniqueness**: SelvaBox spans three neotropical countries, including the region with the world's highest tree species diversity (Amazonian Ecuador), whereas Detectree2 and BCI50ha are geographically isolated. This geographic and ecological diversity fundamentally differentiates SelvaBox from existing tropical datasets.
>
> **Quantified distributional novelty**: To rigorously demonstrate SelvaBox's uniqueness, we conducted a comprehensive analysis of tree crown size distributions using Jensen-Shannon (JS) distance and Kolmogorov-Smirnov (KS) tests. Note that KS test p-values follow $p \approx 2e^{-2 n \cdot D^2}$ [1], where $n$ is dataset size and $D$ the KS statistic, and our dataset sizes range from $n = 3,947$ (Detectree2) to $266,663$ (OAM-TCD), we expect p-values approaching $1.04\cdot 10^{-35}$ when $D \ge 0.1$ and  $n \ge 3,947$. We improved Figure 12 (Appendix F.1) to detail crown-size distributions for each dataset.
>
> SelvaBox exhibits substantially different crown-size distributions from existing tropical datasets (JS Distance: BCI50ha = 0.6248, Detectree2 = 0.6476) and moderate differences from temperate datasets (NeonTreeEvaluation = 0.2789, QuebecTrees = 0.2622). Pairwise KS tests confirm statistically significant differences across all comparisons (p < 0.001, KS values ranging from 0.2117 to 0.6338). These results demonstrate that SelvaBox captures distinct structural and morphological characteristics of tropical forests not represented in existing benchmarks. We add these new results in the revised manuscript in Appendix F.2.
>
> |||BCI50ha|Detectree2|NeonTreeEval.|QuebecTrees|OAM-TCD|
> |-|-|-|-|-|-|-|
> |SelvaBox|JS Distance|0.6248 |0.6476|0.2789|0.2622|0.2152 |
> |SelvaBox| KS Test|0.6231|0.6338 |0.3257|0.2270|0.2117|
> |SelvaBox|KS Test p-value|$<1 \cdot 10^{-35}$|$<1 \cdot 10^{-35}$ |$<1 \cdot 10^{-35}$|$<1 \cdot 10^{-35}$|$<1 \cdot 10^{-35}$|
>
> **Enabling new research directions**: The diversity in crown sizes, shapes, and textures within SelvaBox (see Figure 14 to 16) opens new research opportunities in machine learning and computer vision. Small tree detection remains an open challenge in remote sensing [3, 4, 5] (see L.79), and SelvaBox's scale and diversity enable systematic benchmarking of generalist detection methods in complex environments such as tropical forest data.
>
> **Alignment with ICLR scope**: This work directly addresses ICLR's "sustainability applications" subject area and the "datasets and benchmarks" topics from the call for papers. Forest monitoring of tropical ecosystems is critical: these regions harbor substantial global biodiversity and carbon stocks. SelvaBox provides the foundational resource necessary to accelerate progress in this domain.
>
>
> ### References:
>
> [1] Veitch-Michaelis et al., OAM-TCD: A globally diverse dataset of high-resolution tree cover maps. In NeurIPS 2024.
>
> [2] Marsaglia et al, Evaluating Kolmogorov's Distribution. In Journal of Statistical Software 2003.
>
> [3] Rabbi et al., Detection in Remote Sensing Images with End-to-End Edge-Enhanced GAN and Object Detector Network. In Remote Sensing 2020.
>
> [4] Li et al., Cross-Layer Attention Network for Small Object Detection in Remote Sensing Imagery. In IEEE Journal of Selected Topics in Applied Earth Observations and Remote Sensing 2021.
>
> [5] Bashir & Wang, Small Object Detection in Remote Sensing Images
> with Residual Feature Aggregation-Based Super-Resolution and Object Detector Network. In Remote Sensing 2021.

---

### Official Review · Reviewer_6MAy · 2025-11-02

**Soundness:** 3
**Presentation:** 3
**Contribution:** 3
**Rating:** 8
**Confidence:** 4

**Summary:**

This paper introduces SELVABOX, a large-scale dataset for individual tree crown detection in tropical forests, comprising 83,137 manually annotated bounding boxes from UAV imagery across Brazil, Ecuador, and Panama. The authors conduct extensive benchmarks comparing CNN-based (Faster R-CNN) and transformer-based (DINO) detection methods across multiple resolutions and spatial extents. Key contributions include: (1) the SELVABOX dataset with expert annotations, (2) a comprehensive benchmark including a new raster-level evaluation metric (RF175), (3) demonstration of multi-resolution training for cross-dataset generalization, and (4) open-source tools for preprocessing and inference. The work shows that DINO with Swin-L backbone achieves state-of-the-art performance on both tropical and temperate forest datasets.

**Strengths:**

1. High-quality dataset: Expert annotation by trained biologists (1,284 person-hours) across diverse tropical forest types ensures dataset quality and ecological validity
2. Rigorous experimental methodology: Spatially separated splits, multiple resolutions tested, standard deviations reported, and extensive ablation studies demonstrate scientific rigor
3. Reproducibility: Excellent documentation with code, data, and models publicly released; comprehensive appendices detail preprocessing steps for all datasets
4. Raster-level evaluation: The RF175 metric addresses a real limitation of tile-level metrics for operational forest monitoring applications

**Weaknesses:**

1. Limited technical novelty: The paper is primarily an application/benchmark paper. While valuable, it doesn't introduce novel architectures or training techniques beyond straightforward multi-resolution augmentation
2. Incomplete analysis of RF175 metric. No ablation on the 75% IoU threshold choice
3. Experimental limitations: Missing temporal validation (same site, different time periods)
4. Statistical rigor: Limited discussion of when differences are meaningful

**Questions:**

See above

---

> ### Author Response · Authors · 2025-11-21
>
> We thank the reviewer for the constructive feedback and the positive comments on the dataset quality, experimental rigor, reproducibility efforts, and the RF1_75 metric. Below, we address each concern point by point and indicate the corresponding changes in the revised manuscript (changes are highlighted in blue).
>
> ### W.1. Limited technical novelty:
>
> We acknowledge the reviewer's observation regarding limited architectural novelty. However, this work directly addresses ICLR's "sustainability applications" subject area and the "datasets and benchmarks" topics from the call for papers. Forest monitoring, especially of tropical forests, is a critical research area: tropical forests harbor a significant portion of global forest carbon stocks and biodiversity. Accelerating progress in this domain requires foundational resources, precisely what this work aims to provide.
> Our contributions are threefold:
>
> **First**, we introduce SelvaBox, the largest pantropical forest dataset spanning three countries with 83k annotated trees, an order of magnitude larger than all previous tropical datasets put together. This resource directly addresses a critical data scarcity in forest monitoring and provides a foundation for developing future methods in this domain.
>
> **Second**, we propose a pretrained model that achieves superior generalization by strategically combining multiple high-quality datasets through multi-resolution augmentation. Our evaluation, both in-distribution and out-of-distribution (Tables 4–5), demonstrates consistent performance improvements over established baselines (Detectree2-resize and DeepForest).
>
> **Third**, we establish a robust benchmark that will serve the community. Beyond its value for advancing machine learning and computer vision research, this work provides practitioners with an immediately deployable pretrained model for operational forest monitoring applications.
>
> ### W.2. RF1 ablation study:
>
> We thank the reviewer for raising this point, which was not covered in the first version of the paper. We performed an ablation study on RF1 versus IoU threshold for the four datasets on which RF1 can be defined: SelvaBox, BCI50ha, Detectree2 and QuebecTrees (we have raster-level annotations for these datasets). We used IoU thresholds from 0.50 to 0.95 with a step of 0.05 and produced a plot for each dataset, optimizing the NMS parameters independently for each IoU threshold (see Figure 4 and Appendix E in the revised manuscript).
>
> The RF1 curves for SelvaBox and QuebecTrees are consistent with what we expected from Tables 4 and 5: our DINO-Swin variants significantly outperform competing methods (both Detectree2 checkpoints as well as DeepForest (N)) at every IoU threshold. Interestingly, we find that the Detectree2-resize model performs better at an IoU of 0.70 than at 0.65 on Detectree2, and better than at IoU thresholds in [0.50, 0.65] on BCI50ha. This is not what we initially expected, because the performance of a model should generally decrease as the evaluation IoU threshold becomes more restrictive. However, there are two simple explanations for this: (1) we optimized the NMS parameters independently for each IoU threshold, so the model may perform unusually well at particular thresholds, and (2) the BCI50ha and Detectree2 datasets are much smaller than SelvaBox and QuebecTrees (4.7k and 3.8k annotations compared to 83k and 23k, respectively),  which may result in higher model variance.
>
> Overall, these insights further underline the quality and scale of SelvaBox, compared to smaller datasets where annotation noise and crown size distribution have a larger impact on metric variance. An interesting direction for future work would be to develop an $\text{RF1}\_{50:95}$ metric, analogous to the standard mAP and $\text{mAR}\_{50:95}$, where NMS hyperparameters are optimized against the average RF1 over multiple IoU thresholds on the validation set. We’ve added these new results and limitations to the manuscript in Section 5.3.
>
>
> ### W.3. Temporal validation:
>
> We acknowledge that temporal validation would strengthen model robustness. However, we prioritized spatial diversity over temporal coverage: annotating multitemporal datasets requires repeating the entire annotation procedure at each timestamp as well as including additional labelling efforts on the spatial alignment of each tree, creating a significant trade-off. We chose to maximize ecological diversity across three countries and diverse forest types to enhance generalization capacity, rather than focus on the same sites across time periods. Temporal validation remains valuable future work, particularly for studying tree phenology, that we explore for temporal tree crown segmentation, but is beyond the scope of this study.

---

> ### Author Response · Authors · 2025-11-21
>
> ### W.4. Performance gain quantification:
>
> We thank the Reviewer for suggesting further discussion of differences in model performances. To make the results in Table 4 and 5 more easily understandable by practitioners we propose to report the percentage of $\text{RF1}\_{75}$ (when available, $\text{mAP}\_{50:95}$ otherwise) relative gain of each method compared to the best competing method (among Detectree2 variants and DeepForest (N)) for each dataset. This metric is defined as: Relative Gain=(Method−Best Competing Method)/Best Competing Method x 100, where “Method” corresponds to each row, and “Best Competing Method” corresponds to the competing method reaching the highest RF1 score. Here is an example for the SelvaBox dataset (Table 4), we will update Table 4 and 5 accordingly.
>
> |Method|Train Dataset(s)|SelvaBox RF1 75|SelvaBox OOD|SelvaBox Relative Gain (%)|
> |-|-|-|-|-|
> |Detectree2-resize|D|13.14|✓|0.00 (reference)|
> |Detectree2-flexi |D+urban|9.21|✓|\-29.91|
> |DeepForest|N|6.08|✓|\-53.73|
> |F. R-CNN-ResNet50|N|4.54 (±0.33)|✓|\-65.45|
> |DINO-Swin-L|N|9.94 (±2.12)|✓|\-24.35|
> |DeepForest|S|38.00 (±0.22)|✗|+189.19|
> |F. R-CNN-ResNet50|S|36.37 (±0.37)|✗|+176.79|
> |DINO-Swin-L|S|**48.60 (±0.49)**|✗|**+269.86**|
> |DeepForest|N+Q+O|21.55 (±1.57)|✓|+64|
> |F. R-CNN-ResNet50|N+Q+O|24.77 (±0.38)|✓|+88.51|
> |DINO-Swin-L|N+Q+O|30.81 (±1.53)  |✓|+134.47|
> |DeepForest|N+Q+O+S|35.92 (±1.20)|✗|+173.36|
> |F. R-CNN-ResNet50|N+Q+O+S|30.56 (±1.44)|✗|+132.57|
> |DINO-Swin-L|N+Q+O+S|*47.63 (±0.23)*|✗|*+262.48*|
>
> In addition to our interpretation in Section 5, we will integrate a paragraph “Practical advice” that will clearly state which model would be useful in which context with respect to our expertise and the presented results.

---

### Author Response · Authors · 2025-12-03

We would like to briefly summarize how our rebuttal and revisions address the main concerns raised by the reviewers.

### 1. Contribution and overall assessment
SelvaBox is the largest open-access tropical tree crown detection dataset, with 83k expert-annotated crowns across Brazil, Ecuador and Panama. We propose a new raster-level metric, the RF1_75, and benchmark both CNN and transformer-based detectors. Our multi-resolution, multi-dataset approach achieves state-of-the-art performance on both in-distribution and out-of-distribution settings. Two reviewers assigned a score of 8 (“good paper, accept”), and another increased their score to 8 after our rebuttal and discussion. Across the reviews and discussion, the paper is consistently commended for the quality and uniqueness of the dataset and benchmark. Reviewers also suggested ways to improve the work, such as adding ablation studies and clarifying the evaluation protocol (which we both did), and some raised questions about venue fit. We provide a summary of our rebuttal and discussions below.

### 2. Addressing concerns

**Novelty and statistical analysis (Reviewer QBSE):** We clarified that the paper is submitted under the ICLR “sustainability applications” subject area and “datasets and benchmarks” topic from the call for papers. Beyond scale, we now quantitatively demonstrate that SelvaBox is distributionally distinct from all existing datasets using the Jensen–Shannon distance, KL divergence, and KS tests on crown-size distributions, showing substantial differences to BCI50ha/Detectree2 and moderate but statistically significant differences to temperate datasets. This supports the claim that SelvaBox enables benchmarking in a regime (highly diverse tropical crowns) that is not covered by existing resources.

**RF1_75 and evaluation protocol (Reviewers 6MAy and qCUJ):** We added an RF1 vs IoU ablation (0.50–0.95) on all datasets with raster-level annotations (SelvaBox, BCI50ha, Detectree2, QuebecTrees). Our DINO-Swin models consistently outperform competing methods across all thresholds on the larger datasets, and we discuss the higher variance on small datasets and the role of NMS tuning. We also highlight future work toward an RF1_{50:95}-style metric, analogous to mAP and mAR_{50:95}. Finally, we clarify the split-design pipeline to justify test-set completeness and make the limitations around the Detectree2 split explicit.

**Annotation quality and inter-annotator agreement (Reviewers QBSE, qCUJ and mwqm):** We clarified the two-pass expert annotation and validation protocol (1,284 person-hours, every tile seen at least twice, annotators consulting on difficult cases). While we cannot retrospectively compute inter-annotator metrics due to software limitations, we argue that (1) the cross-dataset generalization results, and (2) consistent performance gains on independent datasets (e.g., QuebecTrees, OAM-TCD) are strong indirect evidence of the high annotation quality. We also expanded the discussion of potential annotation biases and how we mitigated them.

### 3. Updates to the manuscript
We also update the manuscript (changes highlighted in blue) to:

1. Include the RF1–IoU ablation plots and the statistical comparisons of crown-size distributions.

2. Add the relative-gain analysis to the main tables and clearer practical guidance on model choice.

3. Incorporate the additional related-work citations suggested by the reviewers.

4. Add a “Ethical considerations and responsible use” section outlining intended use and potential misuse (e.g., for harmful land exploitation), together with licensing constraints.

Overall, our dataset and benchmark have a clear impact on ecological and tropical forest monitoring, as reflected in the reviews, rebuttal, and revisions. We believe the remaining concerns have been fully addressed through new experiments and additional clarifications in the text.

---

### Meta-Review · Area_Chair_gW2m · 2026-01-06

**Summary:**

This submission introduces SelvaBox, which is a large open-access dataset for tropical tree crown detection from high-resolution UAV RGB orthomosaics, spanning Brazil, Ecuador, and Panama, with ~83k expert-annotated crowns. The paper also provides (i) extensive benchmarks across CNN and transformer detectors, (ii) a raster-level evaluation protocol and metric (RF1_75, raster F1 at IoU≥0.75), and (iii) evidence that higher-resolution inputs and multi-resolution/multi-dataset training improve in-distribution performance and cross-dataset generalisation. The authors additionally emphasise reproducibility via public release of the dataset/code/models.

Across the four reviews, there is strong agreement that the dataset is valuable and unusually large/unique for tropical forests, and that the benchmarking is comprehensive and useful.

Reviewer 6MAy has been strongly positive on dataset quality, rigour, and reproducibility; raised concerns about limited technical novelty lack of RF1_75 threshold ablation, missing temporal validation, and limited discussion of statistical meaningfulness.

Reviewer QBSE: Generally positive on writing/motivation/experiments; key concerns were (a) lack of inter-annotator agreement or direct quality assessment, (b) limited evidence supporting the proposed evaluation protocol and (c) venue-fit/novelty concerns for ICLR. After the rebuttal, they stated they would keep the initial score because inter-annotator analysis remained infeasible.
Reviewer qCUJ (initial score 6 but updated to 8 before the abrupt stop of the discussion period... they raised similar concerns (annotation consistency, test-set completeness, RF1 sensitivity to IoU/NMS, Detectree2 split leakage, need quantitative dataset comparisons, and responsible-use discussion).

Reviewer mwqm has been strongly positive, but noted incomplete annotations in some regions, RGB-only limitations, lack of LiDAR comparison, and potential annotation bias questions; after rebuttal, indicated concerns were addressed and kept rating.

**Reviewer Concerns:**

A couple (addressed) of concerns include the following:

a) RF1_75 justification and sensitivity: a major criticism (6MAy, qCUJ) was the lack of analysis for the IoU=0.75 choice / RF1 sensitivity. The authors added an ablation of RF1 vs IoU thresholds (0.50–0.95), and clarified that NMS is tuned per threshold and that variance is larger on smaller datasets, this materially strengthens the evaluation

b) Test-set completeness and split design: some concerns about test completeness were addressed with a concrete AOI-based splitting pipeline: removing low-quality borders, creating AOI holes where annotations are sparse, selecting the highest-quality regions for the test, and balancing coverage/diversity.

A couple of open concerns are:

a) No inter-annotator agreement metric: Multiple reviewers requested this as direct evidence of label reliability and as an baseline for evaluating metrics. The authors explain that it is infeasible due to annotation tooling/history limitations and timeline constraints. This remains the primary unresolved weakness and is the main reason the most critical reviewer (QBSE) held their score.


b) Limited methodological novelty (by design): The work is primarily a dataset+benchmark contribution rather than proposing a new model.

**Reviewer Scores:**

The reviewers have already given high scores, so I doubt we would have seen any further changes.

---

### Decision · Program_Chairs · 2026-01-26

Accept (Poster)